# EBMDock: Neural Probabilistic Protein-Protein Docking via a Differentiable Energy-based Model

**Huaijin Wu**[1†]**, Wei Liu**[1†]**, Yatao Bian**[2]**, Jiaxiang Wu**[23]**, Nianzu Yang**[1]**, Junchi Yan**[1*]

[1]Department of Computer Science and Engineering, Shanghai Jiao Tong University
[2]Tencent AI Lab  [3]XVERSE, China
{whj1201,captain.130,yangnianzu,yanjunchi}@sjtu.edu.cn
{yatao.bian,jiaxiang.wu.90}@gmail.com
https://github.com/wuhuaijin/EBMDock

## Abstract

Protein complex formation, a pivotal challenge in contemporary biology, has recently gained interest from the machine learning community, particularly concerning protein-ligand docking tasks. In this paper, we delve into the equally crucial but comparatively under-investigated domain of protein-protein docking. Specifically, we propose a geometric deep learning framework, termed EBMDock, which employs statistical potential as its energy function. This approach produces a probability distribution over docking poses, such that the identified docking pose aligns with a minimum point in the energy landscape. We employ a differential algorithm grounded in Langevin dynamics to efficiently sample from the docking pose distribution. Additionally, we incorporate energy-based training using contrastive divergence, enhancing both performance and stability. Empirical results demonstrate that our approach achieves superior performance on two benchmark datasets DIPS and DB5.5. Furthermore, the results suggest EBMDock can serve as an orthogonal enhancement to existing methods.

## 1 Introduction

Protein-protein interactions play a prominent role in diverse biological processes, e.g. signal transduction (Pawson & Nash, 2000), enzymatic reactions (Frieden, 1971; Tetlow et al., 2004), and immune responses (Huang, 2000). Understanding the complex formation between proteins is essential. However, the experimental determination of protein structures is complex and expensive, highlighting the potential advantage of computational approaches in studying protein-protein interactions, especially universal and stable ones in need (Porter et al., 2019). We address the problem of protein-protein docking and follow the common rigid docking setting in recent works (Ganea et al., 2022) without considering the deformations of proteins, which is an approximation for data collection and tractability by existing techniques.

Over the decades, a series of score-based approaches (Schindler et al., 2017; Schneidman-Duhovny et al., 2005; Yan et al., 2017) have emerged for the prediction of complex structures, computationally exploring the binding interactions between proteins, providing insights into protein-protein recognition, binding affinity, and structural rearrangements. These methods typically entail a multi-step procedure as follows: Firstly, they generate numerous complex candidates. Subsequently, a scoring function is employed to rank these candidates (Moal et al., 2013; Basu & Wallner, 2016; Launay et al., 2020; Eismann et al., 2021). Finally, the top-ranked candidates undergo a refinement process utilizing energy or geometric models (Verburgt & Kihara, 2022). As a result, these methods always rely on heavy candidate sampling and a well-tailored score function based on the spatial and chemical properties of every single atom, facing the challenge of sampling efficiency and stability.

---

*Correspondence author. † equal contribution. This work was partly supported by National Key Research and Development Program of China (2020AAA0107600), NSFC (62222607), Tencent AI Lab, and SJTU Trans-med Awards Research (STAR) 20210106. J. Wu's work was done when he was with Tencent AI Lab.

Table 1: Comparison of different deep-learning based (rigid) Protein-Protein docking methods.

| Methods | Formulation | Input | Accuracy | Efficiency | Stability |
|---------|-------------|-------|----------|------------|-----------|
| AF-multimer (Evans et al., 2021) | Iterative search | sequence | High | low | median |
| Equidock (Ganea et al., 2022) | One-shot regression | structure | low | high | high |
| DIFFDOCK-PP (Ketata et al., 2023) | Generative model | structure | median | median | median |
| **EBMDock (Ours)** | Energy model | structure | median | high | high |

On the other hand, there are also emerging lines of research in the area of geometric deep learning and molecular representation learning, and these methods have also been extended to show promise in learning protein structures. Previous works (e.g. EquiDock (Ganea et al., 2022), HMR (Wang et al., 2023)) have approached protein-protein docking by treating it as a regression problem, utilizing Kabsch algorithm (Kabsch, 1976) to align two given proteins and aiming to directly predict the final pose (directly output a rotation matrix $\mathbf{R}$ and a translation vector $\mathbf{t}$). While these methods are efficient, they are limited to generating a single predicted result (one-point solution) without characterizing other possible binding states during the actual docking process. DIFFDOCK-PP (Ketata et al., 2023) formulates docking as a generative problem: given two proteins, it estimates the distribution over all potential poses using a diffusion generative model. The generative model sacrifices efficiency for higher accuracy but lacks stability. Another line of protein structure prediction is AlphaFold(AF) (Jumper et al., 2021), which involves taking the primary sequences of the proteins as input and conducting a search for multiple sequence alignments (MSAs). Although AF-multimer (Evans et al., 2021) achieves high accuracy, it suffers low efficiency and reliance on template features. The comparison of different methods is shown in 1.

In this paper, we advocate for learning a differentiable energy function modeled by geometric deep neural networks. It provides probability distribution over docking poses, based on which we can apply efficient sampling methods to improve the solutions, either from scratch or from a given point solution obtained from an alternative approach e.g. (Ganea et al., 2022; Wang et al., 2023) (using our method as a plug-in), which also enhances the stability. Specifically, we resort to energy-based training methods with Langevin dynamics sampling. This conducts implicit data augmentation, which holds great potential, given the limited availability of existing protein complex structures and their difficulty in obtaining through physical simulations.

We present EBMDock, an energy-based learning framework for generating docking poses represented by SE(3) transformations for protein-protein complexes. EBMDock features a learnable energy function based on distance likelihood, employing geometric deep learning to extract residue features and filter binding interface residues. Inspired by Deepdock (Méndez-Lucio et al., 2021), we model distance distributions using Gaussian mixture models and compute energy as the average negative distance log-likelihoods of binding interface residue pairs. Training involves contrastive divergence and Langevin dynamics for generating negative samples to optimize the energy landscape. In inference, we sample points from the energy distribution using Langevin dynamics, selecting the lowest energy point as the docking pose prediction. Evaluation on the Database of Interacting Proteins (DIPS) and the Docking Benchmarks 5.5 demonstrates that our method surpasses existing deep learning-based approaches and matches the performance of traditional docking software, significantly speeding up the process. Combining EBMDock with EquiDock shows potential for enhancing existing approaches as a plugin. **The highlights of this paper are:**

1) We propose a geometric deep learning-based energy function for (rigid) protein-protein docking which employs statistical potential based on the distance likelihood. This approach formulates the docking problem as an optimization problem, allowing for efficient and stable determination of the optimal docking pose through differentiable sampling methods.

2) The probability distribution over docking poses is derived from the distance distribution and the energy function acts as the measure of confidence, which enables our methods to function as a plugin to boost other one-point solution methods.

3) We propose an energy-based learning framework for the protein docking problem for the first time. Trained with contrastive divergence, which conducts implicit data augmentation, the energy function can overcome data imbalance.

4) Quantitative results show that it outperforms recent deep learning-based models and the speed is 100x faster than traditional docking software. Meanwhile, it performs stably across datasets. Also, empirical results show that it can boost existing deep-learning docking methods by a large margin.

## 2 RELATED WORK

### 2.1 MOLECULAR DOCKING.

Existing molecular docking methods can be categorized into traditional score-based methods and emerging deep learning-based methods. Score-based methods, exemplified by Attract (Schindler et al., 2017), PatchDock (Schneidman-Duhovny et al., 2005) and HDock (Yan et al., 2017), always involve a search algorithm to generate candidates and a scoring function to select (Moal et al., 2013; Basu & Wallner, 2016; Launay et al., 2020; Eismann et al., 2021). Specifically, Attract uses a randomized search algorithm and a physics-based scoring function. The scoring function in PatchDock is based on the geometric shape and electrostatic complementarity of the protein surfaces, while HDock combines both physics-based and empirical terms to estimate the binding energy. However, these methods are all faced with the problems of expensive computation cost and dependence on complex template libraries (Ruiz-Carmona et al., 2014). In contrast, deep learning-based approaches aim to enhance efficiency while maintaining accuracy.

While several methods (Méndez-Lucio et al., 2021; Stärk et al., 2022; Lu et al., 2022; Corso et al., 2023; Zhang et al., 2023) have been developed for protein-ligand docking, they may not directly address the protein-protein docking task investigated in this study. Protein-ligand interactions are often characterized by ligands binding to deep clefts on the protein surface (Vajda & Guarnieri, 2006), whereas protein-protein interactions involve larger surface areas (ranging from 700 to 1500 $Å^2$) and relatively flat binding interfaces (Bahadur et al., 2004; Nooren & Thornton, 2003), posing additional challenges. EquiDock (Ganea et al., 2022) and DIFFDOCK-PP (Ketata et al., 2023) are among the few methods specifically designed for protein-protein docking. In this paper, we pursue the energy-based treatment and propose a learnable energy function for protein-protein docking equipped with binding interface prediction techniques (Sverrisson et al., 2021; Tubiana et al., 2022).

### 2.2 ENERGY-BASED MODELING

Energy-based learning (LeCun et al., 2006) is a learning paradigm that revolves around modeling and learning the underlying distribution of data. Unlike traditional supervised learning where explicit labels are provided in training, energy-based learning focuses on estimating and optimizing the energy associated with different configurations or states of the system and has shown promising results in various domains, including data generation (Du & Mordatch, 2019; Suhail et al., 2021; Xie et al., 2021), out-of-distribution detection (Liu et al., 2020), molecular structure design (Hataya et al., 2021) and combinatorial optimization (Li et al., 2023). There are several systematic approaches to train energy-based models such as score matching (Vincent, 2011) and ratio matching, while gradient-based MCMC methods (Welling & Teh, 2011; Titsias & Dellaportas, 2019) are widely used in inference. This paper employs the contrastive divergence approach (Hinton, 2002) with a Langevin dynamics sampling (Welling & Teh, 2011) method to train the energy function.

## 3 EBMDOCK METHODOLOGY

### 3.1 PRELIMINARIES AND PROBLEM SETUP

We start by giving the formal formulation of the rigid docking problem. We are given an unbounded pair of proteins denoted as ligand (Lig) and receptor (Rec) arbitrarily, with $n$ residues and $m$ residues respectively. For simplicity, we denoted the unbounded pair as $C := (Lig, Rec)$. The locations of residues are represented by the coordinates of its corresponding $\alpha$-carbon atom, denoted as $\mathbf{X}_L \in \mathbb{R}^{3 \times n}$ and $\mathbf{X}_R \in \mathbb{R}^{3 \times m}$. We aim to predict the conformation of their bounded (docked) state with the structure of each part unchanged during the docking process, that is, we view each protein as a rigid body. Keeping the position of receptor fixed, ligand rotate with $\mathbf{R} \in SO(3)$ and translate with $\mathbf{t} \in \mathbb{R}^3$ can lead to a docking pose $(\mathbf{RX}_L + \mathbf{t}, \mathbf{X}_R)$, which means the complex they form is completely determined by a SE(3) transformation applied to the ligand. Following the traditional score-based docking method, we consider that the underlying process of ascertaining $(\mathbf{R}^*, \mathbf{t}^*)$ can be modeled by a score function $F_\theta(\mathbf{R}, \mathbf{t}, C)$ parameterized by $\theta$ and the criteria:

$$\mathbf{R}^*, \mathbf{t}^* = \underset{\mathbf{R} \in SO(3), \mathbf{t} \in \mathbb{R}^3}{\arg\max} F_\theta(\mathbf{R}, \mathbf{t}, C). \tag{1}$$

### 3.2 ENERGY-BASED MODELING FOR RIGID DOCKING

To tackle the optimization problem, i.e. Eq. 1, we first need to construct a proper complex probability density function $p_\theta(\mathbf{R}, \mathbf{t} \mid C)$ positively related to the function $F_\theta(\mathbf{R}, \mathbf{t}, C)$. Here we employ

the energy-based treatment, where $E_\theta(\mathbf{R}, \mathbf{t}, C) = -F_\theta(\mathbf{R}, \mathbf{t}, C)$ and the higher score of function $F_\theta(\mathbf{R}, \mathbf{t}, C)$ represents lower energy:

$$p_\theta(\mathbf{R}, \mathbf{t} \mid C) = \frac{\exp\left(-E_\theta(\mathbf{R}, \mathbf{t}, C)\right)}{Z}, Z := \int \exp(-E_\theta(\mathbf{R}, \mathbf{t}, C)). \qquad (2)$$

Since the observed data only illustrate the ground-truth docking pose, the energy-based modeling is superior due to its maximum entropy (i.e., minimum prior) property (Jeffreys, 1946).

In contrast to the majority of prior research that trains energy models for generative modeling, our focus is to determine the optimal docking pose of a protein pair and its corresponding $(\mathbf{R}, \mathbf{t})$. For this purpose, the decision-making process (inference) of our energy-based model involves comparing the energies associated with various docking poses corresponding to different $(\mathbf{R}, \mathbf{t})$, and selecting the pose with the lowest energy. Therefore, our primary concern is the relative energies, and we can avoid the need to estimate the partition function or compute expectations by training with a carefully designed loss. This allows to parameterize the energy function using any neural architecture, and a thorough discussion on the formulation of the energy loss can be found in (LeCun & Huang, 2005).

## 3.3 ENERGY MODEL ARCHITECTURE

In this section, we will formally introduce our proposed method, which is how to obtain the corresponding energy from a given docking pose that is specific to a protein-protein complex.

**Protein Representation** Proteins consist of several sequences of amino acid residues that fold into 3-dimensional structures in space. In line with Ganea et al. (Ganea et al., 2022), we represent protein as a graph $\mathcal{G} = (\mathcal{V}, \mathcal{E})$. Due to the limitation of computation resources, we work on residue level (our method can be extended to atom level smoothly), which means each node in the graph is an amino acid residue in the protein and each node $i$ has a 3D coordinate $\mathbf{x}_i \in \mathbb{R}^3$ which is the 3D coordinate of $\alpha$-carbon atom of the residue. We use k-nearest-neighbor (KNN) to build the graph of protein and the initial node feature is the embedding of residue type.

**Protein Feature Extractor** After we build the ligand graph $\mathcal{G}_\mathcal{L} = (\mathcal{V}_\mathcal{L}, \mathcal{E}_\mathcal{L})$, receptor graph $\mathcal{G}_\mathcal{R} = (\mathcal{V}_\mathcal{R}, \mathcal{E}_\mathcal{R})$ and get the initial node embedding $\mathbf{F}_L \in \mathbb{R}^{d \times n}, \mathbf{F}_R \in \mathbb{R}^{d \times m}$, we apply a message-passing neural network (MPNN) as our node feature extractor. Here we applied a modified Equiformer as our backbone (Liao & Smidt, 2022), referred to as EquiformerPP. Equiformer leverages the strength of Transformer architectures and incorporates SE(3)-equivariant features, and we add a SE(3)-equivariant cross-attention module to aggregate interactions between the ligand and the receptor. Specifically, the transformation process can be formulated as:

$$\mathbf{Z}_L \in \mathbb{R}^{3 \times n}, \mathbf{H}_L \in \mathbb{R}^{d \times n}, \mathbf{Z}_R \in \mathbb{R}^{3 \times m}, \mathbf{H}_R \in \mathbb{R}^{d \times m} = \mathbf{EquiformerPP}\left(\mathbf{X}_L, \mathbf{F}_L, \mathbf{X}_R, \mathbf{F}_R\right), \qquad (3)$$

where $\mathbf{H}_L, \mathbf{H}_R$ are the output node representations and $d$ is the output dimension. $\mathbf{Z}_L, \mathbf{Z}_R$ are the transformed coordinates of residues but we do not need them in the following.

**Energy based on Distance Likelihood** Potential based on distance likelihood has shown promise in protein-small molecular conformation (Méndez-Lucio et al., 2021). Inspired by that, here we use a deep learning approach to learn a distance distribution that is specific for a node pair (one on ligand and one on receptor), which can be used to compute the energy of a certain docking pose later.

After message passing, we get the output representation of each node, $\{\mathbf{h}_{l,1}, \ldots, \mathbf{h}_{l,n}\}$, $\{\mathbf{h}_{r,1}, \ldots, \mathbf{h}_{r,m}\}$, then we employ a pairwise approach where we concatenate the features of each ligand node with each receptor node. The concatenated features are passed through a mixture density network (MDN) (Bishop, 1994), which comprises a feedforward neural network that generates a set of means $\mu$, standard deviations $\sigma$, and mixing coefficients $\alpha$ to parametrize a Gaussian mixture model for each ligand-receptor node pair. The mixture model depicts the conditional probability density function of distance for each ligand-receptor node pair, enabling us to estimate the likelihood of locating ligand node $i$ separated from a receptor node $j$ by any distance $d_{i,j}$. Specifically, for ligand node feature $\mathbf{h}_{l,i}$ and receptor node feature $\mathbf{h}_{r,j}$ (where $i, j$ means the index of node respectively), the MDN applies an MLP to get a hidden state $\mathbf{h}_{ij}$:

$$\mathbf{h}_{i,j} = \mathrm{MLP}\left(\mathrm{cat}(\mathbf{h}_{l,i}, \mathbf{h}_{r,j})\right). \qquad (4)$$

Then the hidden state is used to calculate the parameters required for the $K = 6$ mixed Gaussians:

$$\mu_{i,j} = \mathrm{ELU}\left(\mathrm{Linear}\left(\mathbf{h}_{i,j}\right)\right) + 1; \qquad (5)$$

$$\sigma_{i,j} = \mathrm{ELU}\left(\mathrm{Linear}\left(\mathbf{h}_{i,j}\right)\right) + 1; \qquad (6)$$

$$\alpha_{i,j} = \mathrm{Softmax}\left(\mathrm{Linear}\left(\mathbf{h}_{i,j}\right)\right). \qquad (7)$$

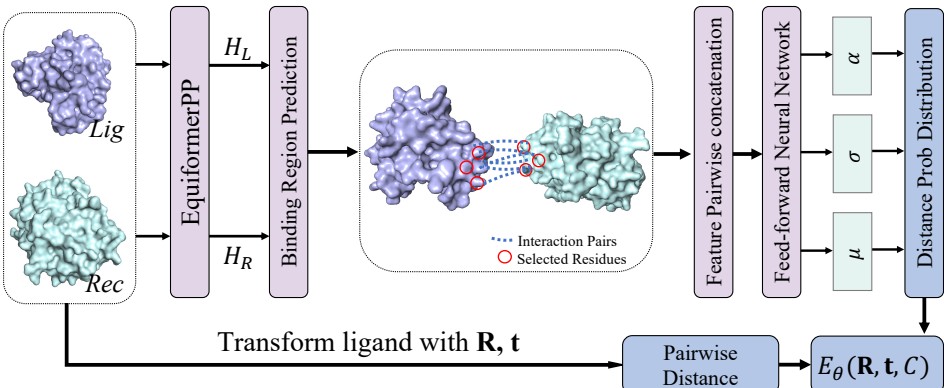

Figure 1: **Construction of EBMDock's energy function**. The ligand and receptor are represented as k-NN graphs where each node is one residue. Features of the residues are extracted from the graphs via EquiformerPP. Based on these features, residues on the binding interface are predicted, and the distance distributions between these residues are modeled with Gaussian mixture models parameterized by $\alpha, \sigma, \mu$. Given a SE(3) transformation $(\mathbf{R}, \mathbf{t})$ of ligand, the pairwise distance between residues can be calculated and the energy is the mean of their negative log-likelihoods.

The negative log-likelihood of distance can be further defined as :

$$-\log P\left(d_{i,j} \mid \mathbf{h}_{l,i}, \mathbf{h}_{r,j}\right) = -\log \sum_{k=1}^{K} \alpha_{i,j,k} \mathcal{N}\left(d_{i,j} \mid \mu_{i,j,k}, \sigma_{i,j,k}\right). \quad (8)$$

Finally, we model the energy as a statistical potential, which is computed by averaging all the independent negative log-likelihood values computed for each ligand-receptor node pair on the binding interface. For any $\mathbf{R} \in SO(3), \mathbf{t} \in \mathbb{R}^3$, we can get the corresponding distance between each node of ligand and receptor, and then we can calculate the energy by:

$$E_\theta(\mathbf{R}, \mathbf{t}, C) = -\frac{1}{|BI|} \sum_{(i,j) \in BI} \log P\left(d_{i,j} \mid \mathbf{h}_{l,i}, \mathbf{h}_{r,j}\right), \quad (9)$$

where $BI$ is the set of residue pairs on the binding interface and $|BI|$ is its cardinality. The docking pose, in which each pair of residues is at the most possible distance from each other, corresponds to the state of minimum energy.

**Binding Interface Prediction** Studies have shown that protein-protein interfaces display comparable chemical and geometric patterns, implying that two proteins may interact if their surfaces exhibit similar shapes and chemical functions (Gainza et al., 2020), thus predicting the functional sites is crucial for comprehending how proteins interact with each other (Sverrisson et al., 2021). Also, when taking the distance between all the receptor and ligand nodes into consideration, it may be greatly affected by the average distance or the centroid distance between the ligand and receptor.

Taking the above factors into account, we first predict the interface where binding might occur, then we only predict the distance distribution of those node pairs on the binding interface. We employ two different approaches, one is predicting the binding site in ligand and receptor respectively, and the other is predicting which pairs of nodes may interact, namely contact prediction. We model both predictions as binary classification problems. To improve the accuracy of the binding interface prediction, we calculate solvent-accessible surface area (SASA) (Lee & Richards, 1971; Mitternacht, 2016) for each protein and incorporate this as an extra feature. See Appendix A for details.

As for the binding site prediction, we classify the nodes in the ligand whose shortest distance to the receptor is smaller than 8Å as positive samples and classify the remaining nodes as negative samples. We do the same for nodes in the receptor. The probability of one ligand node (or receptor node) $v_i$ (or $v_j$) to be on the binding site can be formulated as:

$$p_i = \sigma_S(\text{MLP}(\mathbf{h}_i)), p_j = \sigma_S(\text{MLP}(\mathbf{h}_j)), \quad (10)$$

where MLP denotes a multi-layer perceptron and $\sigma_S$ is implemented as the sigmoid function.

For contact prediction, the node pairs with distance within the threshold 15Å are classified as positive samples. We use a MLP together with a sigmoid to get the contact probability of node $v_i$ and $v_j$:

$$p_{ij} = \sigma_S(\text{MLP}(\text{cat}(\mathbf{h}_i, \mathbf{h}_j))). \tag{11}$$

Note that both classification tasks have label-imbalance issues, thus we apply a weighted focal loss (Qin et al., 2018; Lin et al., 2017) to prevent models from easily classifying all samples as negative ones. The loss function takes the form:

$$\mathcal{L}_{focal}(p) = -\omega(1 - p)^\gamma \log(p), \tag{12}$$

where $\omega$ and $\gamma$ are hyperparameters.

It should be emphasized that both approaches potentially contribute to the identification of the binding interface. Detailed comparisons can be located in Appendix B. There are approaches designed especially to find the protein-protein interaction interface (Tubiana et al., 2022; Sverrisson et al., 2021), we can also utilize their results in our pipeline for energy calculation.

### 3.4 ENERGY-BASED LEARNING FRAMEWORK FOR DOCKING POSE PREDICTION

In this section, we describe our proposed energy-based learning framework for protein-protein rigid docking pose prediction. We take the docking process as an optimization problem by sampling in continuous space which has six degrees of freedom (three of translation and three of rotation), and we use three Euler angles to characterize the rotation specifically.

Given a specific ligand-receptor pair, we use MDN to predict the corresponding distance distribution of each selected node pair (either from binding site prediction or contact prediction), under which we can calculate the energy of a given docking pose, thus explicitly provide the energy distribution of all docking poses. Here, we train our energy model with contrastive divergence:

$$\mathcal{L}_e = E_\theta(\mathbf{R}^+, \mathbf{t}^+, C) - E_\theta(\mathbf{R}^-, \mathbf{t}^-, C), \tag{13}$$

where for a specific protein complex, $(\mathbf{R}^+, \mathbf{t}^+)$ is the ground-truth docking pose and $(\mathbf{R}^-, \mathbf{t}^-)$ is a docking pose sampled from the complex probability density function $p_\theta(\mathbf{R}, \mathbf{t} \mid C)$, which we call negative sample. The negative sample can be generated with Markov chain Monte Carlo (MCMC) sampling. Since our energy function is fully differentiable with respect to $(\mathbf{R}, \mathbf{t})$, we employ Langevin dynamics (Nijkamp et al., 2019; Grathwohl et al., 2019; Xie et al., 2016) as our MCMC transition kernel following recent developments in Energy-Based Models (EBMs). Specifically, we start from a random sampled docking pose $(\mathbf{R}^0, \mathbf{t}^0)$, and perform:

$$\mathbf{R}^{\tau+1} = \mathbf{R}^\tau - \frac{\lambda}{2}\nabla_{\mathbf{R}} E_\theta(\mathbf{R}, \mathbf{t}, C) + \epsilon^\tau, \tag{14}$$

$$\mathbf{t}^{\tau+1} = \mathbf{t}^\tau - \frac{\lambda}{2}\nabla_{\mathbf{t}} E_\theta(\mathbf{R}, \mathbf{t}, C) + \epsilon^\tau, \tag{15}$$

where $(\mathbf{R}^\tau, \mathbf{t}^\tau)$ represents the docking pose after $\tau$ iterations and $\epsilon^\tau$ is sampled from a normal distribution $\mathcal{N}(0, \lambda)$. The process is similar to gradient descent with an added Gaussian noise and we can arrive at a low-energy docking pose through a series of steps. It is worth noting that the energy is a statistical potential based on the distance likelihood and no neural network parameters are involved in energy computation from $(\mathbf{R}, \mathbf{t})$. Given a pair of unbounded ligand and receptor, we can get the predicted distance distribution based on only *one* forward propagation of the geometric neural network. Thus given a docking pose $(\mathbf{R}, \mathbf{t})$, we can compute the distance log-likelihoods quickly which yield the energy and its derivatives. This means our iterative Langevin dynamics sampling is very efficient in practice.

### 3.5 ENERGY-BASED TRAINING AND INFERENCE

The overall training loss is composed of three parts, including the contrastive divergence $\mathcal{L}_e$, the L2 regularization loss $\mathcal{L}_r$ on energy values, and weighted focal loss $\mathcal{L}_{focal}(\cdot)$ of binding interface prediction. The L2 regularization loss $\mathcal{L}_r = E_\theta(\mathbf{R}^+, \mathbf{t}^+, C)^2 + E_\theta(\mathbf{R}^-, \mathbf{t}^-, C)^2$ is used to avoid the energy values get arbitrarily large causing gradient overflow problem. The total loss is:

$$\mathcal{L}_{total} = \mathcal{L}_e + \lambda_1 \mathcal{L}_r + \lambda_2 \mathcal{L}_{focal}(\cdot), \tag{16}$$

Table 2: Statistics of two datasets for experiments.

| Items | DIPS-Het | DB5.5 |
|---|---|---|
| avg. # of residues per receptor / ligand | 259.4 | 265.1 |
| avg. # of residues on binding pocket per receptor / ligand | 25.5 | 26.9 |
| avg. # of pocket residue pairs per complex | 1079.5 | 779.8 |
| #train / #valid / #test | 9,093 / 1,146 / 1,153 | — / — / 253 |

where $\lambda_1$ and $\lambda_2$ are hyperparameters. The energy-based training is shown in Fig. 2.

For inference, we add an intersection loss $\mathcal{L}_{IS}(\mathbf{R}, \mathbf{t}, C)$ as part of the energy to avoid atom clash, so the inference energy is:

$$E_{infer} = E_\theta + \alpha \mathcal{L}_{IS}, \qquad (17)$$

where $\alpha$ is a hyperparameter (see Appendix C for details). Then we start from random $(\mathbf{R}^0, \mathbf{t}^0)$ and use the Langevin dynamics sampling which approximately solves the problem in an iterative manner to find $(\mathbf{R}^*, \mathbf{t}^*)$ corresponded to the lowest energy. The iteration formula is the same as Eq. 14 & 15, and we will take enough steps till convergence.

As the node features from EquiformerPP satisfy SE(3) equivariance, and the energy calculation process as well as the Langevin dynamics sampling also exhibits equivariance, our framework maintains SE(3) equivariance.

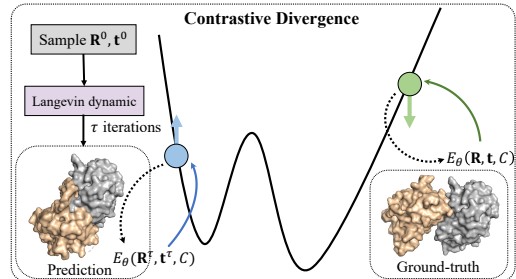

Figure 2: Energy-based training. EBMs are trained with contrastive divergence, where the energy of the ground truth complex (green dot) decreases and the energy of the sampled complex (blue dot) increases. Blue/green arrows indicate forward computation while dashed arrows indicate gradient back-propagation.

## 4 EXPERIMENTS

In this section, we conduct experiments to comprehensively evaluate our EBMDock. All deep learning-based methods run on a machine with i9-10920X CPU, RTX 3090 GPU, and 128G RAM.

### 4.1 EXPERIMENTAL SETTING

**Datasets** We conduct numerical experiments on two datasets: Database of Interacting Protein Structures (DIPS) (Townshend et al., 2019) and Docking Benchmarks 5.5 (DB5.5) (Vreven et al., 2015). DIPS is a comprehensive dataset of protein complex structures, sourced from the Protein Data Bank (Berman et al., 2000) and DB5.5 is a benchmark dataset renowned for its high-quality data manually curated by domain experts. Similar to (Wang et al., 2023), we collect a modified version of DIPS, referred to as DIPS-Het, to focus on interactions between different proteins. Both datasets are tailored for rigid body docking. The statistics of datasets are demonstrated in Table 2.

**Baselines** We compare our EBMDock method with several state-of-the-art docking methods, including score-based docking software HDock (Yan et al., 2017), PatchDock (Schneidman-Duhovny et al., 2005) as well as deep learning-based algorithm EquiDock (Ganea et al., 2022), DIFFDOCK-PP (Ketata et al., 2023) and AlphaFold-Multimer (Evans et al., 2021). We test HDock with the local packages they provided and test PatchDock on the webserver. We test AlphaFold-Multimer through the colab interface they provided. For DIffDOCK-PP, we use their best DIPS-validated model in all related experiments. It should be noticed that HDock, PatchDock and AlphaFold-Multimer use different training data compared to our models. They might have utilized parts of our test sets for extracting templates or as training examples, which may lead to an optimistic result.

**Evaluation Metrics** To measure the quality of the predicted docking pose, we report Complex Root Mean Square Deviation (CRMSD), Interface Root Mean Square Deviation (IRMSD), as well as the overall quality of docking evaluation metric DockQ (Basu & Wallner, 2016). For CRMSD, we use the Kabsch algorithm (Kabsch, 1976) to superimpose the predicted and ground-truth complex before computing the RMSD between them. To compute IRMSD, we employ a similar approach but only use the coordinates of the interface residues with a distance of less than 8Å to the other protein's residues in the ground-truth complex. DockQ is computed following the settings in (Basu

Table 3: **Rigid protein-protein docking results on 100 samples from the DIPS-Het test set.** The number of initial poses sampled from the diffusion or energy model is in parentheses. EBMDock-interface means that we assume the binding interface is given. Note that DIFFDOCK-PP fails[2]on some samples (6 out of 100). The results we presented are obtained after removing these outliers.

| Metric | Complex RMSD ↓ | | | Interface RMSD ↓ | | | DockQ ↑ | | | Inference |
|---|---|---|---|---|---|---|---|---|---|---|
| Methods | Mean | Median | std | Mean | Median | std | Mean | Median | std | Time (sec) |
| PatchDock | 19.34 | 17.95 | 10.30 | 17.16 | 16.17 | 10.35 | 0.04 | 0.02 | 0.20 | 2232 |
| HDock | 2.04 | 0.23 | 5.98 | 2.67 | 0.25 | 8.51 | 0.88 | 0.98 | 0.28 | 782 |
| AlphaFold-Multimer | 6.45 | 4.66 | 8.17 | 6.14 | 2.12 | 8.43 | 0.48 | 0.52 | 0.31 | 1940 |
| EquiDock | 11.73 | 10.94 | 7.19 | 11.43 | 10.82 | 6.60 | 0.12 | 0.05 | 0.18 | 3.9 |
| DIFFDOCK-PP(5) | 12.09 | 11.87 | 9.88 | 13.79 | 12.50 | 11.72 | 0.24 | 0.04 | 0.35 | 30.9 |
| **EquiDock+plugin(1)** | 10.16 | 9.13 | 8.09 | 10.23 | 9.13 | 8.08 | 0.17 | 0.08 | 0.20 | 5.0 |
| **EBMDock (5)** | 9.09 | 7.95 | 6.83 | 8.98 | 6.93 | 6.20 | 0.22 | 0.12 | 0.22 | 8.3 |
| **EBMDock(5)-interface** | 2.89 | 1.56 | 3.97 | 2.05 | 1.46 | 1.54 | 0.63 | 0.68 | 0.76 | 8.5 |

Table 4: **Rigid protein-protein docking results on DB5.5.** EBMDock* means using another interface prediction method to roughly find the interaction interface. Note that DIFFDOCK-PP fails on some samples (13 out of 253). The results we presented are obtained after removing these outliers.

| Metric | Complex RMSD ↓ | | | Interface RMSD ↓ | | | DockQ ↑ | | | Inference |
|---|---|---|---|---|---|---|---|---|---|---|
| Methods | Mean | Median | std | Mean | Median | std | Mean | Median | std | Time (sec) |
| PatchDock | 16.46 | 16.52 | 7.56 | 15.60 | 15.29 | 6.03 | 0.05 | 0.02 | 0.01 | 951 |
| HDock | 5.55 | 0.42 | 9.42 | 5.19 | 0.31 | 8.99 | 0.72 | 0.97 | 0.42 | 703 |
| AlphaFold-Multimer | 7.65 | 4.86 | 8.03 | 6.41 | 1.69 | 7.66 | 0.48 | 0.55 | 0.10 | 2540 |
| EquiDock | 17.15 | 15.90 | 5.31 | 14.56 | 14.10 | 4.91 | 0.03 | 0.02 | 0.04 | 3.8 |
| DIFFDOCK-PP(5) | 17.56 | 17.21 | 7.87 | 17.76 | 17.12 | 8.47 | 0.04 | 0.01 | 0.09 | 37.3 |
| **EquiDock+plugin(1)** | 16.78 | 15.94 | 5.11 | 13.76 | 12.73 | 4.56 | 0.05 | 0.02 | 0.04 | 4.9 |
| **EBMDock (5)** | 14.79 | 15.64 | 5.05 | 12.47 | 10.99 | 5.06 | 0.05 | 0.04 | 0.04 | 7.4 |
| **EBMDock*(5)** | 12.61 | 13.26 | 6.52 | 10.27 | 11.56 | 5.70 | 0.16 | 0.05 | 0.20 | 10.4 |
| **EBMDock(5)-interface** | 3.71 | 2.52 | 3.84 | 2.25 | 1.82 | 1.48 | 0.57 | 0.582 | 0.23 | 7.6 |

& Wallner, 2016). For a fair comparison, we utilize only the $\alpha$-carbon coordinates for computing all the metrics. In order to have a direct comparison of efficiency, we also report the inference time.

**Training and Inference Details** Our models are trained and evaluated on the training and validation set of DIPS-Het, respectively. As for the training loss $\mathcal{L}_{total}$, we set $\lambda_1$ and $\lambda_2$ as 0.1 and 50. When computing the contrastive divergence $\mathcal{L}_e$, we randomly sample three groups of $(\mathbf{R}_i^0, \mathbf{t}_i^0)$, 50 steps of Langevin dynamics are then applied to them independently and the average of three energies serves as the second term in Eq. 13 to ensure the stability of training. We use Adam with learning rate 3e-4 and weight decay 1e-4 as the optimizer. More implementation details can be found in Appendix F.

We evaluate on the full DB5.5 and 100 samples randomly selected from the DIPS-Het test set, we provide the pdb id of the chosen 100 samples in Appendix F. Again, we randomly sample 5 groups of $(\mathbf{R}_i^0, \mathbf{t}_i^0)$ and use Langevin dynamics sampling with 100 steps. The $(\mathbf{R}_i^{100}, \mathbf{t}_i^{100})$ with the lowest energy is the predicted SE(3) transformation. In addition, we use a deep learning-based method EquiDock to generate an initial $(\mathbf{R}^0, \mathbf{t}^0)$ for each complex and optimize it with our energy function and Langevin dynamics sampling. Since other methods themselves are time-consuming, here we only use our method as a plug-in to boost EquiDock. We also try to equip our method with another interface prediction method on DB5.5, utilizing their results in our pipeline for energy calculation.

## 4.2 RESULTS OF DOCKING POSE GENERATION

Results are shown in Table 3 & 4 and Fig. 4 & 5. It can be observed that EBMDock consistently outperforms EquiDock and PatchDock across both datasets. Compared to DIFFDOCK-PP, EBMDock demonstrates superior performance across various metrics, except for a slight disadvantage in the average DockQ metric. The results show that both EBMDock and EquiDock achieve very low standard deviations, indicating that EBMDock exhibits stability compared to all other methods. Utilizing an existing interface prediction method (Tubiana et al., 2022) can further enhance our results, named as **EBMDock*(5)**. Providing the ground-truth binding interface, EBMDock consistently achieves favorable conformation of the complex and delivers competitive results compared to HDock, named as **EBMDock(5)-interface**. These results indicate that when we have a rough knowledge of the

---

[2]We consider cases with CRMSD metric exceeding 50Å as failure cases.

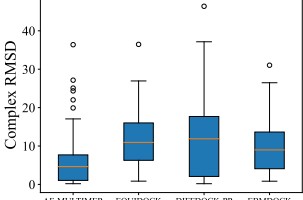 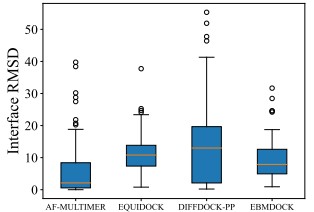 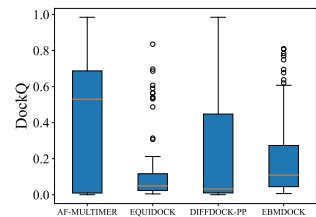

Figure 4: Distributions of three evaluation metrics on DIPS-Het test set (100 selected samples). Here we choose to compare the results of the deep learning-based method.

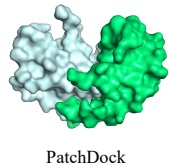 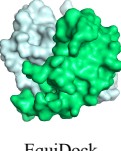 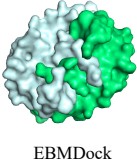 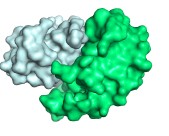 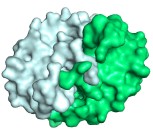

| PatchDock | EquiDock | EBMDock | Equidock with Boost | |
|---|---|---|---|---|
| DockQ:0.047 | DockQ:0.038 | DockQ:0.559 | DockQ:0.187 | Ground Truth |

Figure 5: Visualization of the protein complex 2ebg, which is successfully predicted by EBMDock.

interaction interface, our energy function and sampling technique exhibit remarkable effectiveness. Additionally, when taking the predicted docking poses of EquiDock as the initial points, EBMDock can improve the poses by moving them to the local minimum nearby, leading to a large improvement beyond EquiDock, named as **EquiDock+plugin(1)**. In terms of speed, EBMDock is 100 times faster than traditional software and AF-multimer, and also nearly 5 times faster than DIFFDOCK-PP. This result demonstrates the efficiency of our sampling approach and also aligns with the statements mentioned in Section 3.4. More experiment results are shown in Appendix E.

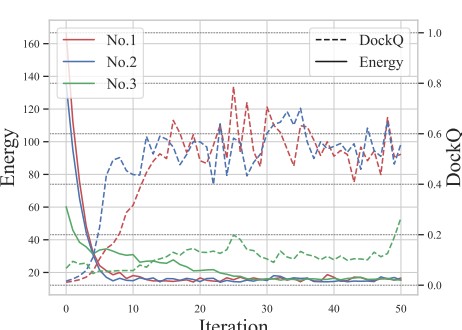

Figure 3: Sampling the top 3 candidates with Langevin dynamics for a protein complex.

To further showcase sampling from the probability distribution, here we show the Langevin dynamics sampling process of the top 3 candidates for one chosen protein complex in Fig. 3. For each candidate, the DockQ increases as the energy decreases, indicating the effectiveness of our energy function and Langevin dynamics sampling. However, a low energy solution does not necessarily correspond to a high DockQ because the energy landscape may have more than one local optimum, some of which may be far from the ground truth, as the 'No.3' candidate solution shows: even though the 'No.3' candidate just has slightly higher energy than the 'No.2' candidate after 50 iterations, it might correspond to a local optimum that is far away from the global optimum.

Moreover, compared with HDock, PathDock, and AF-multimer, we only consider the residue types and coordinates as the initial features and do not rely on any finely designed task-specific features.

## 5 CONCLUSION

We have presented EBMDock, an energy-based learning framework for protein-protein docking. By leveraging advanced geometric neural networks and distance likelihood modeling, it provides an energy probabilistic distribution over docking poses, enabling efficient sampling for improved solutions. The framework incorporates energy-based training with Langevin dynamics sampling and demonstrates superior performance compared to existing deep learning-based methods. EBMDock also serves as a plugin to enhance the capabilities of other docking methods. Future work will explore multiple protein docking which involves graph matching methods for multiple graphs (Jiang et al., 2021).

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

# A    SURFACE FEATURE

This section introduces the method we use to calculate surface-aware node features. We use solvent-accessible surface area (SASA) (Lee & Richards, 1971; Mitternacht, 2016) to represent the surface degree of each residue. Specifically, solvent-accessible surface area (SASA) measures the surface area of a molecule that is accessible to solvent molecules. It provides valuable information about the molecular shape, volume, and interactions with the surrounding environment. SASA can be calculated using various algorithms. Here we adopt the Shrake-Rupley (Shrake & Rupley, 1973) algorithm and the general steps involved are as follows:

- Define the probe radius, denoted as $r_{probe}$.
- Generate the solvent-accessible surface by placing a probe sphere of radius $r_{probe}$ at discrete points on the molecular surface. Determine if each point is accessible to the probe molecule, considering no overlap with the molecule.
- Calculate the surface area of each atom using the area of its van der Waals sphere, denoted as $A_i$. Calculate the overlapping area with adjacent atom $j$'s van der Waals spheres, denoted as $A_{ij}$.
- Compute the SASA for each atom $i$ by $\text{SASA}_i = A_i - \sum_j A_{ij}$. Then sum up the SASA values for all atoms within the residue to obtain the SASA value of each residue.

We then normalize the calculated SASA values to a number between $0$ and $1$ to represent the surface degree of each residue, a higher value indicates a greater surface degree. The normalization is performed by dividing all SASA values in a protein by the largest one in it. We analyze the frequency of unnormalized and normalized SASA values across all proteins in DB5.5, as shown in Fig. S6. The SASA values of residues on the binding interface are also analyzed for comparison. It can be observed that the distribution of SASA values of residues on the binding interface is different from that of all residues, indicating that this feature can be helpful for binding interface prediction.

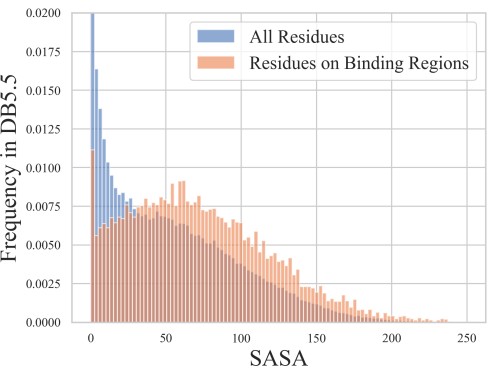 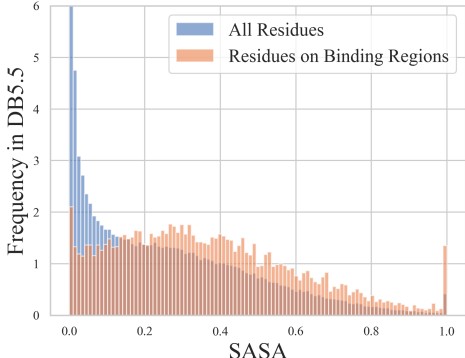

Figure S6: Distribution of unnormalized (left) and normalized (right) residue SASA across all proteins in DB5.5. Residues on the binding interface are analyzed specially to be compared with all residues.

# B    BINDING SITE PREDICTION VS CONTACT PREDICTION

In Section 3.3, we present two approaches for predicting the binding interface in protein complexes: binding site prediction (BSP) and contact prediction (CP). BSP involves predicting the binding sites on the ligand and receptor respectively, while CP focuses on predicting the residue pairs that interact with each other. To construct the energy function, we select the relevant residue pairs based on the prediction results and consider only those pairs in the calculation. For BSP, we use the Cartesian product of the predicted binding sites on the ligand and receptor to identify the interacting residue pairs. On the other hand, CP directly utilizes the predicted residue pairs without further modification.

During the training phase, we employ BSP and CP respectively to compute the loss for binding interface prediction. Through experimentation, we observe that training the energy models with BSP

loss as an auxiliary loss results in stabler performances. During the inference phase, we evaluate the performance of BSP and CP on the DIPS-Het test set. The results, presented in Table S5, demonstrate that CP outperforms BSP across all evaluation metrics. This indicates that CP is capable of selecting high-quality residue pairs with better generalization capabilities.

Table S5: Comparison on DIPS-Het of different binding interface prediction methods. BSP means binding site prediction, CP means contact prediction. AUC (area under the receiver operating characteristic curve) and AP (average precision) are metrics of the binding interface prediction module.

| Method | AUC ↑ | AP ↑ | Complex RMSD ↓ | | Interface RMSD ↓ | | DockQ ↑ | |
|---|---|---|---|---|---|---|---|---|
| | | | Mean | Median | Mean | Median | Mean | Median |
| BSP | 0.86 | 0.67 | 11.49 | 10.23 | 11.13 | 9.46 | 0.17 | 0.09 |
| **CP** | 0.95 | 0.58 | **9.74** | **8.98** | **9.23** | **7.90** | **0.20** | **0.11** |

## C  INTERSECTION LOSS

In a natural protein complex, the ligand and receptor should never intersect with each other. In our energy function, however, the optimum points may lead to varying degrees of intersection. To alleviate this issue, we add an auxiliary term to the energy function during inference—the intersection loss $\mathcal{L}_{IS}(\mathbf{R}, \mathbf{t}, C)$. Following previous work (Sverrisson et al., 2021; Ganea et al., 2022), the surface of a protein point cloud $\mathbf{X} \in \mathbb{R}^{3 \times n}$ is expressed as $\{\mathbf{x} \in \mathbb{R}^3 : G(\mathbf{x}) = \gamma\}$, where $G(\mathbf{x}) = -\sigma \ln \left( \sum_{i=1}^{n} \exp \left( -\|\mathbf{x} - \mathbf{x}_i\|^2 / \sigma \right) \right)$. For a point $\mathbf{x} \in \mathbb{R}^3$ outside the protein, it has $G(\mathbf{x}) > \gamma$. To avoid intersections in the complex, each residue in the ligand should be outside the receptor, so as to the residues in the receptor. As a result, we can write the intersection loss as:

$$\mathcal{L}_{IS}(\mathbf{R}, \mathbf{t}, C) = \frac{1}{n} \sum_{i=1}^{n} \max(0, \gamma - G_R(\mathbf{R}\mathbf{x}_{Li} + \mathbf{t})) + \frac{1}{m} \sum_{j=1}^{m} \max(0, \gamma - G_{L(\mathbf{R},\mathbf{t})}(\mathbf{x}_{Rj})), \quad (18)$$

where $G_R$ is the surface of receptor and $G_{L(\mathbf{R},\mathbf{t})}$ is the surface of ligand after transformed with $(\mathbf{R}, \mathbf{t})$. We set the parameters $\gamma = 8$ and $\sigma = 8$ so that the ground truth has $\mathcal{L}_{IS}(\mathbf{R}^*, \mathbf{t}^*, C) = 0$ and no "holes" are inside a protein. Finally, the energy function for inference is $E_{infer}(\mathbf{R}, \mathbf{t}, C) = E_\theta(\mathbf{R}, \mathbf{t}, C) + \alpha \mathcal{L}_{IS}(\mathbf{R}, \mathbf{t}, C)$. $\alpha$ is a parameter that controls the influence of intersection loss on energy function and partly depends on the shape of the interface in the complex. We find $\alpha = 1.5$ works well on DB5.5 and $\alpha = 0.5$ works well on DIPS-Het. We conduct experiments on the DIPS-Het test set with different $\alpha$ and the results are shown in Table S6.

Table S6: Docking Performance on DIPS-Het with different coefficient $\alpha$ of $\mathcal{L}_{IS}$.

| $\alpha$ value | Complex RMSD ↓ | | Interface RMSD ↓ | | DockQ ↑ | |
|---|---|---|---|---|---|---|
| | Mean | Median | Mean | Median | Mean | Median |
| 0 | 9.74 | 8.98 | 9.23 | 7.90 | 0.20 | 0.11 |
| **0.5** | **9.09** | **7.95** | **8.98** | **6.93** | **0.22** | **0.12** |
| 5 | 9.66 | 9.13 | 10.16 | 9.67 | 0.19 | 0.08 |
| 50 | 11.63 | 11.40 | 13.00 | 12.41 | 0.08 | 0.04 |

## D  ADDITIONAL EXPERIMENTS

### D.1  EXPERIMENTS ON ANTIBODY-ANTIGEN DOCKING

We have claimed that HDock and AlphaFold-Multimer use different training data compared to our models. They might have utilized parts of our test sets for extracting templates or as training examples, which may lead to an optimistic result. For a fair comparison, we follow (Luo et al., 2023) and select 68 antibody-antigen samples from the Protein Data Bank which were released after October 2022 and have not been used to train AlphaFold-Multimer or as templates in HDOCK. The PDB

Table S7: Ablation studies on DIPS-Het.

| Binding Interface Prediction | CD Loss | Complex RMSD ↓ | | Interface RMSD ↓ | | DockQ ↑ | |
|---|---|---|---|---|---|---|---|
| | | Mean | Median | Mean | Median | Mean | Median |
| — | — | 18.87 | 18.50 | 19.47 | 19.63 | 0.03 | 0.01 |
| ✓ | — | 13.34 | 14.94 | 10.55 | 11.02 | 0.14 | 0.05 |
| ✓ | ✓ | 9.74 | 8.98 | 9.23 | 7.90 | 0.20 | 0.11 |

Table S8: **Rigid protein-protein docking results on 68 antibody-antigen samples.** The number of initial poses sampled from the diffusion or energy model is in parentheses. EBMDock-interface means that we assume the binding interface is given. Note that DIFFDOCK-PP fails on 4 samples (Complex RMSD metric exceeds 50 Å). The results we presented are obtained after removing these outliers.

| Metric | Complex RMSD ↓ | | DockQ ↑ | |
|---|---|---|---|---|
| Methods | Mean | std | Mean | std |
| HDock | 15.78 | 6.36 | 0.09 | 0.19 |
| AlphaFold-Multimer | 13.65 | 5.89 | 0.11 | 0.17 |
| EquiDock | 18.47 | 2.71 | 0.04 | 0.02 |
| DIFFDOCK-PP(5) | 23.82 | 7.04 | 0.02 | 0.06 |
| **EBMDock (5)** | 14.87 | 4.93 | 0.05 | 0.02 |
| **EBMDock(5)-interface** | 6.51 | 5.13 | 0.32 | 0.22 |

id of these samples are listed here: 8dls, 8dlr, 8dfi, 8dfh, 8dcc, 8dad, 7zr8, 7zf8, 7xxl, 7xh8, 7x26, 7wsl, 7wsi, 7ws6, 7ws2, 7wrz, 7wrv, 7wro, 7wrl, 7wrj, 7wog, 7wlc, 7wef, 7wee, 7wed, 7wcr, 7wbz, 7urq, 7uaq, 7tty, 7ttx, 7ttm, 7tpj, 7tp4, 7tp3, 7tlz, 7the, 7tc9, 7t8w, 7t7b, 7t01, 7swp, 7su1, 7str, 7sem, 7sd5, 7sbu, 7sbg, 7sbd, 7sa6, 7s5p, 7rxp, 7rxi, 7rbu, 7qtk, 7n0a, 7lo8, 7lo7, 7kql, 7fjc, 7f7e, 7f6z, 7f6y, 7eng, 7ek0, 7ejz, 7ejy, 7e9p.

The results are shown in Table S8. Compared to Table 3 and 4, both HDOCK and AlphaFold-Multimer exhibit a significant decrease in performance on unseen cases. Although EBMDOCK's results also decline, they are close to those of AF-Multimer and surpass HDOCK in CRMSD. Additionally, when EBMDOCK is able to obtain rough docking interface information, the docking results are far superior to both AF-Multimer and HDOCK.

## D.2 EXPERIMENTS ON FULL DIPS-HET TEST SET

In Table 3, we report the results on only 100 samples from the DIPS-Het test set. That's because methods like PatchDock, HDock, and AlphaFold-Multimer have too long inference time. These baselines limit us from conducting a fair comparison on a more extensive dataset. We have shown that our method is highly efficient and relatively stable, allowing scalability to large-scale datasets. Here, we present our results on the full DIPS-Het test set (1153 samples) in Table S9.

## E ABLATION STUDY

We conduct additional experiments on DIPS-Het to further examine the contributions of each proposed technique. The model without binding interface prediction uses all ligand-receptor node pairs to calculate energy, while the model without contrastive divergence simply minimizes the energy of ground-truth complexes. The results in Table S7 show that both techniques improve the performances. Specifically, the binding interface prediction module mainly improves the Interface RMSD since it can find the residues on the protein-protein interfaces, while contrastive divergence contributes to the overall performance, which is in line with our design purposes.

Table S9: **Rigid protein-protein docking results on full DIPS-Het test set.** The number of initial poses sampled from the energy model is in parentheses. EBMDock-interface means that we assume the binding interface is given.

| Metric | Complex RMSD ↓ | | | Interface RMSD ↓ | | | DockQ ↑ | | |
|---|---|---|---|---|---|---|---|---|---|
| Methods | Mean | Median | std | Mean | Median | std | Mean | Median | std |
| EBMDock (5) | 9.18 | 8.12 | 6.23 | 9.12 | 7.23 | 5.94 | 0.21 | 0.11 | 0.20 |
| EBMDock(5)-interface | 2.93 | 1.65 | 3.87 | 2.14 | 1.53 | 1.54 | 0.62 | 0.68 | 0.71 |

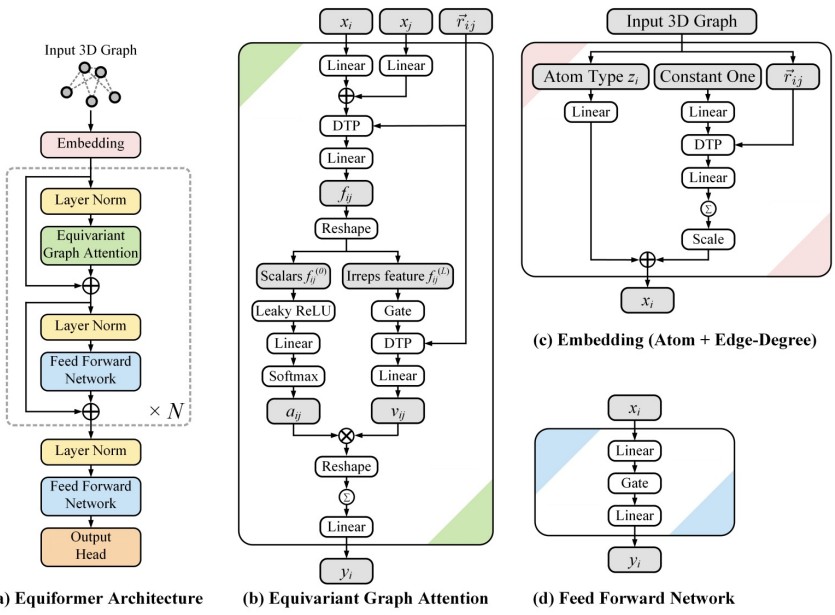

(a) Equiformer Architecture  (b) Equivariant Graph Attention  (c) Embedding (Atom + Edge-Degree)  (d) Feed Forward Network

Figure S7: Architecture of Equiformer (Liao & Smidt, 2022). Equiformer embeds input 3D graphs with atom and edge-degree embeddings and processes them with Transformer blocks, consisting of equivariant graph attention and feed-forward networks. "$\otimes$" denotes multiplication, "$\oplus$" denotes addition, and "DTP" stands for depth-wise tensor product. $\sum$ within a circle denotes summation over all neighbors. Gray cells indicate intermediate irreps features.

## F  IMPLEMENTATION DETAILS

### F.1  THE ARCHITECTURE OF EQUIFORMERPP

The origin architecture of Equiformer has been shown in Fig. S7, which is directly extracted from the original paper Liao & Smidt (2022). Since the Equiformer is designed for one 3D graph, not capturing any interaction of two graphs, we add a SE(3)-equivariant cross-attention module to aggregate interactions between the ligand and the receptor. Here we will describe this cross-attention module, shown in Fig. S8.

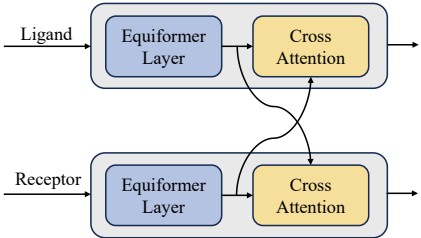

Figure S8: Demonstration of the cross-attention module.

Given the ligand features $y_L$ and receptor features $y_R$, the cross-attention layer enables communication between proteins:

$$y'_L = Gate(\frac{(y_L W_Q)(y_R W_K)^T}{\sqrt{d}})(y_R W_V),$$

$$y'_R = Gate(\frac{(y_R W_Q)(y_L W_K)^T}{\sqrt{d}})(y_L W_V),$$

where $y_L \in \mathbb{R}^{n \times d}, y_G \in \mathbb{R}^{m \times d}$ denotes the features of the ligand/receptor protein, $d$ denotes the dimension of features, $n/m$ denotes the number of nodes in the ligand/receptor graphs. $W_Q, W_K$, and $W_V$ are the parameter matrices for the query, key, and value in attention computation, respectively. The gate is an equivariant activation function, which can be found in the original paper Liao & Smidt (2022). The modified architecture is named EquiformerPP.

## F.2    TEST SAMPLES

Here we provide the pdb id of the selected 100 samples: 6R17, 5W3X, 6J4O, 5CK3, 2V4I, 1M57, 2A4R, 5JH5, 1MU2, 7QIH, 5A63, 7ATE, 4ZXS, 4XSS, 4H62, 7RNW, 4OZ1, 5MR3, 6B1U, 4Y5O, 7ZVY, 2HHF, 6FC1, 7VKB, 4LN0, 3ZV0, 7SZ0, 1YRT, 1JEQ, 2D5R, 3BPQ, 5HQP, 4EEC, 6CWX, 1EGP, 3G5O, 2P1O, 1QOV, 3SND, 4DFC, 1E1H, 4IYP, 4HRT, 3L4Q, 5ZWB, 5YRH, 4Y61, 1TY4, 1LPH, 4J2L, 5ZWL, 5D7G, 6NVW, 3OJY, 2FBW, 5JKC, 7U7N, 7MU2, 2F8V, 6C0Y, 7NNL, 5BRK, 3MEU, 5H67, 4UE6, 6ZDX, 6U07, 5QZU, 5IC6, 7N1N, 5DOB, 4OYD, 3C5W, 6GZC, 7LTS, 3A5Z, 7KM6, 5ME5, 2DSP, 4I5N, 3SXU, 5M5E, 6KMQ, 1JLT, 7F5M, 5VLL, 6FTO, 4QLB, 5KY7, 3KYS, 7CNR.

## F.3    EVALUATION METRICS

We have briefly described the evaluation metrics in Section 4.1, and we will give further explanations here. We assess the performance of rigid protein docking using three metrics: Complex RMSD, Interface RMSD, and DockQ. Let's denote the ground truth and predicted complex structures as $\mathbf{Z}^* \in \mathbb{R}^{3 \times (n+m)}$ and $\mathbf{Z} \in \mathbb{R}^{3 \times (n+m)}$, respectively. When calculating the Complex RMSD (C-RMSD), we first align $\mathbf{Z}$ with $\mathbf{Z}^*$ using the Kabsch algorithm and then use the formula $\sqrt{\frac{1}{n+m}|\mathbf{Z}^* - \mathbf{Z}|_F^2}$, where $|\cdot|_F$ denotes the Frobenius norm. For the calculation of Interface RMSD (IRMSD), we follow a similar approach but only consider the coordinates of the interface residues of the ground truth complex. These residues are defined as those within a distance of 8Å from the other protein's residues. DockQ is based on three standardized criteria from the Critical Assessment of Predicted Interactions (CAPRI): $L_{rms}$, $I_{rms}$, and $f_{nat}$. To compute $L_{rms}$, we first superimpose the backbone atoms of the receptors in the ground truth and predicted complex, and then calculate the ligand RMSD based on the coordinates of the backbone atoms of corresponding ligands. For $I_{rms}$, we align the interface residues (residues with $\alpha$-carbon within a distance of 8 Å from any $\alpha$-carbon in the other protein) and calculate the backbone RMSD. Lastly, $f_{nat}$ represents the recall in recovering residue-residue contacts between the proteins, where two residues are considered "in contact" if their $\alpha$-carbon atoms have a distance less than 8 Å. DockQ is a continuous score ranging from 0 to 1, derived from $L_{rms}$, $I_{rms}$, and $f_{nat}$. A higher DockQ score indicates better performance in terms of docking predictions. Note that *backbone* means that all atoms in residues are used for calculation, not only the $\alpha$-carbon atoms.

## F.4    TRAINING DETAILS

Here we provide more details about the training process. We first preprocess the protein complexes by randomly assigning the roles of ligand and receptor and randomly translating and rotating the ligand while keeping the receptor fixed, which are used to examine the SE(3)-equivariance of our method. The calculation of the energy function only considers the residue pairs on the binding interface. Note that the information about the binding interface comes from ground truth during training, and binding site prediction or contact prediction during inference. When computing the contrastive divergence $\mathcal{L}_e$, we randomly sample three groups of $(\mathbf{R}_i^0, \mathbf{t}_i^0)$ from the distribution of $U(-\pi, \pi)$ and $N(\mu, \sigma)$, respectively, where $\mu$ is a vector from the center of the ligand to the center of receptor and $\sigma$ is the average distance between two centers in complexes, set as 40Å in our

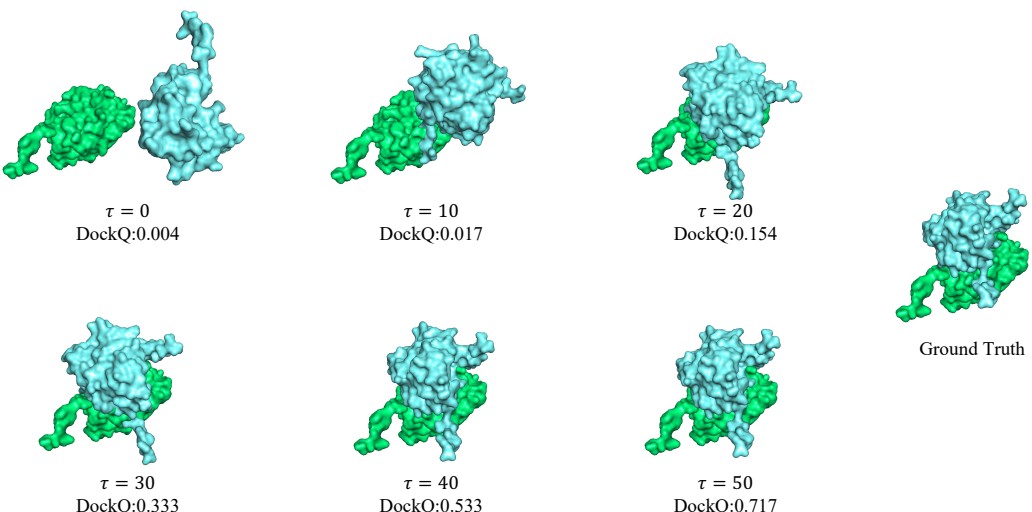

Figure S9: Visualization of Langevin dynamics sampling process of the protein complex 1a95.

experiments. Then $\tau$ steps of Langevin dynamics are applied to them independently and the average of three energies serves as the second term in Eq. 13. Besides, we set bounds for $\mathbf{R}$ and $\mathbf{t}$. For $\mathbf{R}$, the bounds are $[-\pi, \pi]$, and values out of the range are handled by adding $2k\pi$, where $k$ is a proper integer. For $\mathbf{t}$, the bounds are $[-400, 400]$, a big enough range for protein complexes, and values out of the range are simply clamped. As for the training loss $\mathcal{L}_{total}$, we observe that the orders of magnitude of three terms are 1, 10, and 1e-3, respectively. Accordingly, we set $\lambda_1$ and $\lambda_2$ as 0.1 and 50, respectively. More hyperparameters are demonstrated in Table S10. We train EBMDock on a machine with i9-10920X CPU, RTX 3090 GPU, and 128G RAM for 3 days.

Table S10: Hyperparameter choices of EBMDock and the training phase settings.

| | Hyperparameters | Values |
|---|---|---|
| **Network** | Node degree (for k-NN) | 10 |
| | Dimension of $h_i$ of EquiformerPP layers | 128 |
| | Number of Mixed Gaussians | 6 |
| | Dropout Rate | 0.2 |
| | Binding Interface Prediction Focal Loss Weight $\omega$ | 0.25 |
| | Binding Interface Prediction Focal Loss Gamma $\gamma$ | 2 |
| **LD** | Learning Rate of Langevin dynamics($\frac{\lambda}{2}$) | 0.01 |
| | Number of Samples of Langevin dynamics | 3 |
| | Number of Steps of Langevin dynamics ($\tau$) | 50 |
| **Training** | Batch Size | 4 |
| | # Epoches | 50 |
| | Optimizer | Adam |
| | Learning Rate | 3e-4 |
| | Weight Decay | 1e-4 |

## G    MORE VISUALIZATION

To show the effectiveness of Langevin dynamics sampling, we present a visual example of protein complex 1a95 in Fig. S9. As the iterations $\tau$ increase, the predicted complexes have higher DockQ and are closer to the ground truth.

## H    DISCUSSION

In this section, we will discuss EBMDock and the possible improvements in future work further.

Firstly, EBMDock uses residue-level modeling but can be easily extended to the atom level. Since internal structures of residues of the same class are relatively similar and modeling on residue level can be computationally efficient, we represent the residues with their alpha carbon. To extend EBM-Dock to the atom level, we only need to modify the nodes in the protein graph from representing a residue to an atom. Instead of learning the statistical potential of residue distances, we would learn the statistical potential of atomic distances. The rest of the network structure and training methods remain unchanged. However, the computing time and memory consumption will increase a lot. If we generalize our methods to flexible docking, which is very challenging, we can still use residue-level modeling and rely on existing tools such as Rosetta to complete the side chain conformation based on the predicted backbone structures.

Secondly, we believe energy-based modeling is promising for the protein-protein docking problem. By carefully designing the deep learning-based energy function, energy-based modeling has clear physical meaning while maintaining high inference efficiency. The AF-Multimer method, which is based on multiple sequence alignments (MSAs), is difficult to improve in terms of efficiency and has significant limitations when dealing with large-scale data. Although the accuracy of energy-based modeling cannot match that of AF-Multimer currently, it has the potential to be used for large-scale virtual screening. Besides, the performances of EBMDock are limited by the scarcity of protein structural information while the protein sequence information is easily accessible and abundant. Given sufficient time, we can leverage AF to predict complex structures and generate new training data to further expand our approach.

Finally, the proposed energy-based learning framework has good extensibility. The backbone can be replaced by new representation learning methods such as those based on the protein surface. The energy function can be generalized to contain more terms. As long as an energy term is differentiable, it can be incorporated into our framework easily. For example, the HDOCK and AF-Multimer achieve good performance on old proteins because they directly use information from training sets such as templates and sequence alignments. We can design new differentiable energy terms to measure how well the docking poses conform to the templates. We can also design energy terms based on physics, for example, van der Waals dispersion/repulsion, directional hydrogen bond energy, Coulomb electrostatic energy, and desolvation-free energy. As the calculation of these terms may be expensive, we can approximate them using deep learning methods. We believe they can improve the performance and will explore these possibilities in future work.

## I    RELATED WORK OF GEOMETRIC DEEP LEARNING

Graph structured data is a ubiquitous type of data and it finds particular resonance within the realm of computational biology (Zhang et al., 2021; Yang et al., 2022a). Notably, complex biological entities, ranging from molecules to proteins, can all be represented through graphs. Moreover, Graph Neural Networks (GNNs), a method commonly used for graph structures, have been widely applied in computational biology (Xu et al., 2019; Guo et al., 2020; Yang et al., 2022b; 2023). Unlike generic graph data, molecular systems in 3D space exhibit symmetries, such as translations and rotations (Satorras et al., 2021; Gasteiger et al., 2021). Recently, many studies have incorporated such inductive biases into GNN design. These novel methods have demonstrated superior performance in tasks including molecular property prediction (Liu et al., 2022), conformation generation (Hoogeboom et al., 2022), particle system dynamics (Satorras et al., 2021), and conformation-based energy estimation (Rezende et al., 2019).

