# OpenReview forum: "EBMDock: Neural Probabilistic Protein-Protein Docking via a Differentiable Energy Model"
_ICLR.cc/2024/Conference — ICLR 2024 poster_

### Official Review · Reviewer_94kF · 2023-10-29

**Soundness:** 3 good
**Presentation:** 3 good
**Contribution:** 2 fair
**Rating:** 6
**Confidence:** 4

**Summary:**

The paper proposes EBMDock, an energy-based learning framework for rigid protein-protein docking. It models the docking problem as optimizing the pose that minimizes the energy function.  The energy function is a statistical potential based on the distance likelihood between residue pairs on the binding interface. It is modeled using Gaussian mixture models parameterized by a geometric deep neural network. EBMDock is trained with contrastive divergence using Langevin dynamics for negative sampling. This allows implicit data augmentation to handle limited protein complex data.

**Strengths:**

- The energy-based formulation provides a nice probabilistic framework for modeling the docking problem and enables efficient sampling methods. Contrastive divergence training is clever for data augmentation.
- The method maintains equivariance throughout the pipeline leveraging geometric deep learning. This is important for generalizability.
- Ablation studies analyzing the impact of different binding site prediction methods add nicely to the empirical analysis.

**Weaknesses:**

- The empirical results of the proposed method are not as strong as expected, as it fails to outperform traditional methods and AlphaFold-multimer. This raises questions about the effectiveness of the proposed approach in comparison to existing techniques.

- The use of only 100 samples for the DIPS-Het test set is questionable. It is unclear if there are any constraints that prevent the application of the proposed method to larger datasets. The authors should provide more information on this aspect and, if possible, test their method on a more extensive dataset.

- The omission of xTrimoDock [1] from the baselines is a notable oversight. According to [1], xTrimoDock can outperform both HDock and AlphaFold-Multimer, and it is also relatively fast. The authors should consider including xTrimoDock in their comparison to provide a more comprehensive evaluation of their proposed method.

- While the performance of EBMDock-interface is impressive, it relies on the ground-truth binding interface, which might not be a fair comparison to other baselines. This could lead to misunderstandings regarding the method's actual performance. I suggest that the authors clarify this point and consider presenting the results of EBMDock-interface separately in a standalone table to avoid potential confusion.

[1] xTrimoDock: Rigid Protein Docking via Cross-Modal Representation Learning and Spectral Algorithm

**Questions:**

- It is intriguing to observe that AF-Multimer outperforms rigid PP docking, especially considering that AF2-Multimer relies solely on sequence inputs and does not incorporate any structural information. Could the author provide an explanation for this phenomena?
- While the energy function is modeled as statistical potential using distance likelihoods, the formulation seems rather simplistic. Could more complex energy formulations further improve accuracy?
- For the MDN, why choose a 6-component GMM? Is performance sensitive to the number of mixtures?
- How was the EquiformerPP architecture designed? What modifications were made to Equiformer?
- How does inference time scale with protein complex size? Could efficiency be an issue for very large complexes?
- Have the authors considered extending this to flexible protein docking? What changes would be needed?

---

> ### Author Response · Authors · 2023-11-17
>
> Thank you for your time and valuable feedback. Below is the detailed response.
>
> > ***Q1: "The empirical results of the proposed method are not as strong as expected, as it fails to outperform traditional methods and AlphaFold-multimer. This raises questions about the effectiveness of the proposed approach in comparison to existing techniques."***
>
> The AlphaFold method, which is based on multiple sequence alignments (MSAs), is difficult to improve in terms of efficiency and cannot be used for large-scale virtual screening. Even if their performance is good, there are significant limitations when dealing with large-scale data. Our approach aims to improve accuracy while maintaining high inference efficiency, although there is currently no method that can achieve this effectively. Therefore, even though our work may not match AlphaFold's, our work has made progress and provided new insights for researchers.
>
> > ***Q2: "The use of only 100 samples for the DIPS-Het test set is questionable. It is unclear if there are any constraints that prevent the application of the proposed method to larger datasets. The authors should provide more information on this aspect and, if possible, test their method on a more extensive dataset."***
>
> We select 100 samples for testing because methods like PatchDock, HDock, and AlphaFold-Multimer have too long inference time. These baselines limit us from conducting a fair comparison on a more extensive dataset. We have shown that our method is highly efficient and relatively stable, allowing scalability to large-scale datasets in our paper. Here, we present our results on the entire DIPS-Het test set (1153 samples). We apologize for the misunderstanding caused and hope that our clarification can alleviate your concerns about our method.
>
> | Method  | CRMSD ($\downarrow$) mean/median/std  | IRMSD ($\downarrow$)  mean/median/std |   DockQ ($\uparrow$) mean/median/std |
> | -- | -- | -- | -- |
> | EBMDock(5)  | 9.18/8.12/6.23 | 9.12/7.23/5.94 |   0.21/0.11/0.20  |
> | EBMDock(5)-interface | 2.93/1.65/3.87 | 2.14/1.53/1.54 |  0.62/0.68/0.71 |
>
> Rigid protein-protein docking results on samples from the DIPS-Het test set.
>
> > ***Q3: "The omission of xTrimoDock[1] from the baselines is a notable oversight. According to [1], xTrimoDock can outperform both HDock and AlphaFold-Multimer, and it is also relatively fast. The authors should consider including xTrimoDock in their comparison to provide a more comprehensive evaluation of their proposed method."***
>
> We conduct comparative testing on the antibody-antigen dataset released in the xTrimoDock[1]. Unfortunately, we are unable to access their publicly available code, which results in differences in the training data and may lead to some unfair comparisons. Since we can not reproduce their experiments, the results related to xTrimoDock[1] in the table are directly extracted from their paper. Also, the inference time is not listed due to the different machines we used and it's not reproducible.
>
> Results on antibody-antigen dataset released in xTrimoDock  (68 samples). Note that DIFFDOCK-PP fails on 4 samples (CRMSD metric exceeds 50Å). The results we presented are obtained after removing these outliers.
>
> | Method      | CRMSD ($\downarrow$) | DockQ ($\uparrow$) |
> | ----------- | -------------------- | ------------------ |
> | HDOCK                |      15.779±6.364                |           0.090±0.187         |
> | AF-Multier           |        13.650±5.886              |             0.108±0.172       |
> | EquiDock             |     18.468±2.706                 |           0.043±0.017          |
> | xTrimoDock |   10.090±7.817            |        0.220±0.232            |
> | DIFFDOCK-PP(5)       |   23.817±7.041       |   0.018±0.064           |
> | EBMDOCK(5)           |       14.870±4.930              |            0.054±0.022        |
> | EBMDOCK(5)-interface |      6.511±5.126              |           0.324±0.218       |
>
> > ***Q4: "While the performance of EBMDock-interface is impressive, it relies on the ground-truth binding interface, which might not be a fair comparison to other baselines. This could lead to misunderstandings regarding the method's actual performance. I suggest that the authors clarify this point and consider presenting the results of EBMDock-interface separately in a standalone table to avoid potential confusion."***
>
> Thanks for your nice suggestion, we will modify this part to avoid confusion. Furthermore, the highlight of our work lies in proposing and utilizing learnable energy for docking, as well as further integrating EBM training and Langevin dynamics sampling for structure prediction. We have not dedicated our efforts to enhancing the interface recognition algorithm. Therefore, we provide results under the assumption that we have a rough knowledge of the interaction interface.
>
> [1] Luo, Yujie, et al. "xTrimoDock: Rigid Protein Docking via Cross-Modal Representation Learning and Spectral Algorithm." bioRxiv (2023): 2023-02.

---

> > ### Author Response · Authors · 2023-11-17
> >
> > > ***Q5: "It is intriguing to observe that AF-Multimer outperforms rigid PP docking, especially considering that AF2-Multimer relies solely on sequence inputs and does not incorporate any structural information. Could the author provide an explanation for this phenomenon?"***
> >
> > AlphaFold-Multimer demonstrates higher accuracy compared to structure-based protein complex prediction methods. Here are some possible reasons that can explain this phenomenon:
> >
> > + AlphaFold has been trained on a large-scale dataset, while our method's training set only contains around 10,000 data points. AlphaFold benefits from a larger training scale, which allows it to learn more robust representations of protein structures.
> >
> > + AlphaFold utilizes sequence homology and evolutionary information for structure prediction. By analyzing a vast amount of protein sequences, the model can identify sequence patterns and co-evolutionary signals that are crucial for understanding protein structure and interactions.
> >
> > + AlphaFold requires extensive time for inference to search for homologous information and optimize the structure. Moreover, as the sequence length increases, the inference time and optimization difficulty for AlphaFold also significantly increase.
> >
> > In summary, the higher accuracy of AlphaFold-Multimer compared to structure-based methods may be explained by its larger training scale, utilization of sequence homology and evolutionary information, as well as the time-consuming nature of its inference and structure optimization processes, which become more challenging with longer sequences.
> >
> > > ***Q6: "While the energy function is modeled as statistical potential using distance likelihoods, the formulation seems rather simplistic. Could more complex energy formulations further improve accuracy?"***
> >
> > Our energy-based learning framework has good extensibility. As long as an energy term is differentiable, it can be incorporated into our framework easily. Through the experimental results, we find that the HDOCK and AF-Multimer achieve good performance on old proteins because they directly use information from training sets such as templates and sequence alignments. We can design new differentiable energy terms to measure how well the docking poses conform to the templates. We can also design energy terms based on physics, for example, van der Waals dispersion/repulsion, directional hydrogen bond energy, Coulomb electrostatic energy, and desolvation free energy. The calculation of these terms may be expensive, but we can approximate them using deep learning methods. Although these terms may increase the computational cost, we believe they can improve the performance. We will explore these possibilities in future work.
> >
> > > ***Q7: "For the MDN, why choose a 6-component GMM? Is performance sensitive to the number of mixtures?"***
> >
> > The performance is not that sensitive to the number of mixtures, we have tried the number range from 4 to 10. We simply take an intermediate value, so we set the number to 6.
> >
> > > ***Q8: "How was the EquiformerPP architecture designed? What modifications were made to Equiformer?"***
> >
> > We are sorry that we have not elaborated on the architecture of EquiformerPP. We add this part in Appendix E.1, see new pdf for details.
> >
> > > ***Q9: "How does inference time scale with protein complex size? Could efficiency be an issue for very large complexes?"***
> >
> > The inference time of EBMDock is composed of two parts, the construction of energy function through geometric deep learning and the sampling process from the distribution via Langevin dynamics. The first part is influenced by the protein complex size, but it takes up only a small part of the inference time even when the complex has more than 2000 residues. The second part only considers the residues on the binding interface, which don't increase significantly as the complex size increases. Besides, the calculation time for this part is linear to the number of residues on the binding interface. As a result, the overall inference time is mainly determined by the step of Langevin dynamics sampling. Previous methods such as AF-Multimer, however, are greatly influenced by the complex size.
> >
> > > ***Q10: "Have the authors considered extending this to flexible protein docking? What changes would be needed?"***
> >
> > Flexible docking poses a greater challenge compared to rigid docking, and as far as our knowledge extends, there are currently no machine learning-based approaches capable of effectively addressing the complexities inherent in flexible protein docking problems, especially rigid docking remains an area of extensive research.
> >
> > To further extend our method to flexible docking, we need additional degrees of freedom, such as incorporating torsion angle sampling on local protein scaffolds. If so, the complexity of the sampling process would intensify, thereby augmenting the difficulties associated with optimization.

---

> > > ### Comment · Reviewer_94kF · 2023-11-22
> > > **Response to the Author Rebuttal**
> > >
> > > Thank you so much for the detailed responses. Part of my concerns are addressed, and I am happy to raise my score.

---

> > > > ### Author Response · Authors · 2023-11-22
> > > >
> > > > Thank you for acknowledging our work. Your rating has provided us with great encouragement and we will continue to refine our paper to make it even better.

---

### Official Review · Reviewer_AgXx · 2023-10-31

**Soundness:** 3 good
**Presentation:** 3 good
**Contribution:** 3 good
**Rating:** 8
**Confidence:** 3

**Summary:**

The authors propose EBMDock, an energy-based approach to rigid protein-protein docking. They sample from the distribution of docking poses using the energy function and show competitive performance compared to other ML and classical docking methods. The EBMDock inference time is substantially less than more computationally intensive methods like AlphaFold-Multimer and PatchDock. EBMDock’s flexibility allows for using the method with initial guesses from other methods like EquiDock.

**Strengths:**

The method is well-motivated and provides a nice contrast to prevailing ML methods in this area. The focus on inference time performance is practical and helpful for practitioners.

**Weaknesses:**

It is still broadly unclear if ML methods for protein-protein docking have any utility compared to "classical" methods. This is still an interesting area of research, with much further progress to be made, but clear discussion of limitations and future directions is particularly warranted.

**Questions:**

1. What is the performance on binding site and contact prediction, before pose prediction and docking? How does this performance compare to binding site and contract prediction classifiers?

2. Do the authors have any ideas about what explains the performance gap to HDock and how they could improve the EBMDock protocol? The results suggest that, based on DockQ scores, DiffDock-PP, EquiDock, and EBMDock without providing the ground truth binding site are not usable for pose prediction. This is a valuable result, but it also motivates proposals for substantial improvements.

3. It would be helpful to define all of the variations (Equidock+plugin, EBMDock (5), etc.) in a single place before the results are presented. The naming conventions could also be made clearer and more specific.

---

> ### Author Response · Authors · 2023-11-17
>
> Thank you for acknowledging our work. Your rating has provided us with great encouragement, and your feedback has been immensely helpful. Below is our detailed response.
>
> > ***Q1: "What is the performance on binding site and contact prediction, before pose prediction and docking? How does this performance compare to binding site and contract prediction classifiers?"***
>
> |  | AUC | AP |
> | --- | --- | --- |
> | BSP |  0.86   |  0.67   |
> | CP    |   0.95  |   0.58  |
>
> As for the performance of the binding site and contact prediction, the details can be seen in Appendix B. As for other methods typically designed for predicting the binding site of the interface, such as SCANNET、MASIF, we are sorry for not being able to compare with them fairly. Since they only consider the binding sites on one protein, not considering the docking counterpart.
>
> > ***Q2: "Do the authors have any ideas about what explains the performance gap to HDock and how they could improve the EBMDock protocol? The results suggest that, based on DockQ scores, DiffDock-PP, EquiDock, and EBMDock without providing the ground truth binding site are not usable for pose prediction. This is a valuable result, but it also motivates proposals for substantial improvements."***
>
> HDock, PatchDock, and AlphaFold-Multimer use different training data compared to our models. They might have utilized parts of our test sets for extracting templates or as training samples, which may lead to the optimistic results. Therefore, we select 68 antibody-antigen samples from the Protein Data Bank which were released after October 2022 and have not been used to train AlphaFold-Multimer or as templates in HDOCK. The result here demonstrates that both HDOCK and AF-Multimer exhibit a significant decrease in performance on unseen cases. Although EBMDOCK's results also decline, they are close to those of AF-Multimer and surpass HDOCK in CRMSD. Additionally, when EBMDOCK is able to obtain rough docking interface information, the docking results are far superior to both AF-Multimer and HDOCK.
>
> | Method    | CRMSD ($\downarrow$) | DockQ ($\uparrow$) |
> | -------------------- | -------------------- | ------------------ |
> | HDOCK   |  15.779±6.364    |   0.090±0.187   |
> | AF-Multier  |   13.650±5.886     |   0.108±0.172       |
> | EBMDOCK(5)   |    14.870±4.930     |   0.054±0.022        |
> | EBMDOCK(5)-interface |      6.511±5.126     |  0.324±0.218       |
>
> We acknowledge that the results are currently not usable, but the protein-protein docking is still a problem far from being solved. Our energy-based learning framework can be improved in many aspects.
>
> The HDOCK and AF-Multimer achieve good performance on old proteins because they directly use information from training sets such as templates and sequence alignments. We can design new differentiable energy terms to measure how well the docking poses conform to the templates.  We can also design energy terms based on physics, for example, van der Waals dispersion/repulsion, directional hydrogen bond energy, Coulomb electrostatic energy, and desolvation free energy. The calculation of these terms may be expensive, but we can approximate them using deep learning methods. They can be incorporated into our framework easily. Besides, our backbone can be easily replaced by new representation learning methods such as those based on the protein surface. For a particular category of docking problems such as antibody-antigen docking, we can train EBMDock on these proteins to learn a more refined energy function, leading to a usable model. We will explore these possibilities in future work.
>
> > ***Q3: "It would be helpful to define all of the variations (Equidock+plugin, EBMDock (5), etc.) in a single place before the results are presented. The naming conventions could also be made clearer and more specific."***
>
> Thanks for your nice suggestion, we will add a table of the naming convention. The highlight of our work lies in proposing and utilizing learnable energy for docking, as well as further integrating EBM training and Langevin dynamics sampling for structure prediction. We have not dedicated our efforts to enhancing the interface recognition algorithm. Therefore, we provide results under the assumption that we have a rough knowledge of the interaction interface. During inference, the number of poses sampled from the energy model is in parentheses.
>
> | Name        | Meaning     |
> | -------------   | ------------------------------------------------------------------------------------------------- |
> | Equidock+plugin(1)   | Take the predicted docking poses of EquiDock as the initial point of EBMDock, use the energy function as a plugin. |
> | Equidock(5)     |  Know nothing about the interaction interface, use CP or BSP module to predict. |
> | Equidock*(5)    | Use another interface prediction method to roughly find the interaction interface, use 5 samples. |
> | Equidock(5)-interface| Given the ground truth of interaction interface. |

---

> > ### Comment · Reviewer_AgXx · 2023-11-21
> >
> > I have read the rebuttal and appreciate the authors' efforts. I am happy to increase my score. I would encourage the authors to emphasize the points related to performance on the temporal split in the paper. Broadly speaking, all deep learning approaches for docking have failure modes in real-world applications, and I think exploring new ideas (as the authors have done in this paper) and being forthcoming about performance is needed for the field to progress.

---

> > > ### Author Response · Authors · 2023-11-22
> > >
> > > Thank you very much for your recognition and suggestions. We will continuously improve the paper, and we will emphasize the point you mentioned.

---

### Official Review · Reviewer_XEYz · 2023-10-31

**Soundness:** 2 fair
**Presentation:** 3 good
**Contribution:** 2 fair
**Rating:** 3
**Confidence:** 4

**Summary:**

This paper proposes an energy-based model, called EBMDock, for protein-protein docking. EBMDock learns an energy function $E(R, t)$, where $R$ is a rotation and $t$ is a translation. This energy function is parameterized by a modified Equiformer architecture and trained using contrastive divergence, a standard algorithm for EBM optimization. At test time, EBMDock uses langevin dynamics to find the optimal rotation and translation (the lowest energy). EBMDock is evaluated on standard protein docking benchmarks and compared with a list of prior methods for protein docking.

**Strengths:**

* Formulating protein docking as an energy-based model is an interesting idea.
* The presentation of the method is relative clear.

**Weaknesses:**

* The performance of EBMDock is much worse than AlphaFold-multimer (AFM) (DockQ: 0.16 vs 0.48). Although EBMDock runs much faster than AFM, the performance is far from AFM. As a result, EBMDock is not going to be useful in practical setting since its prediction is mostly inaccurate.
* EBMDock is only evaluated on rigid-body docking setting, where the ground truth structure of each protein is given. Therefore, its comparison with AFM is not fair because it does not assume the binding site is known and it folds both proteins from sequence directly.
* In the case where ground truth is not available, we still need to use AF2 to predict the structure of each protein. As a result, the entire workflow for EBMDock won't be much faster than AFM.
* Gradient of rotation $\nabla_R E(R, t)$ is not defined clearly.
* Equiformer-PP architecture is not described.

**Questions:**

* In practice, how do you calculate the gradient $\nabla_R E(R, t)$?

---

> ### Author Response · Authors · 2023-11-17
>
> Thank you for your time and valuable feedback. Below is the detailed response.
>
> > ***Q1: "The performance of EBMDock is much worse than AlphaFold-multimer (AFM) (DockQ: 0.16 vs 0.48). Although EBMDock runs much faster than AFM, the performance is far from AFM. As a result, EBMDock is not going to be useful in practical setting since its prediction is mostly inaccurate."***
>
> The AlphaFold method, which is based on multiple sequence alignments (MSAs), is difficult to improve in terms of efficiency and cannot be used for large-scale virtual screening. Even if their performance is good, there are significant limitations when dealing with large-scale data. Our approach aims to improve accuracy while maintaining high inference efficiency. Even though the performance may not match that of AlphaFold, our work has made progress and provided new insights for researchers.
>
> Also, we have claimed that HDock, PatchDock, and AlphaFold-Multimer use different training data compared to our models. They might have utilized parts of our test sets for extracting templates or as training examples, which may lead to an optimistic result. Therefore, we select 68 antibody-antigen samples from the Protein Data Bank which were released after October 2022 and have not been used to train AlphaFold-Multimer or as templates in HDOCK. The result here demonstrates that both HDOCK and AF-Multimer exhibit a significant decrease in performance on unseen cases. Although EBMDOCK's results also decline, they are close to those of AF-Multimer and surpass HDOCK in CRMSD. Additionally, when EBMDOCK is able to obtain rough docking interface information, the docking results are far superior to both AF-Multimer and HDOCK. We will add these results to our manuscript and hope they can ease your concerns.
>
> | Method               | CRMSD ($\downarrow$) | DockQ ($\uparrow$) |
> | -------------------- | -------------------- | ------------------ |
> | HDOCK                |      15.779±6.364                |           0.090±0.187         |
> | AF-Multier           |        13.650±5.886              |             0.108±0.172       |
> | EBMDOCK(5)           |       14.870±4.930             |            0.054±0.022        |
> | EBMDOCK(5)-interface |      6.511±5.126              |           0.324±0.218       |
>
> > ***Q2: "EBMDock is only evaluated on rigid-body docking setting, where the ground truth structure of each protein is given. Therefore, its comparison with AFM is not fair because it does not assume the binding site is known and it folds both proteins from sequence directly."***
>
> We acknowledge that there may be some unfair comparisons when comparing with AlphaFold, but this is due to the inherent differences in our settings. In fact, other structure-based prediction methods also struggle to make a fair comparison with AlphaFold. When it comes to protein docking, we usually have known monomer structures and predict the resulting conformation after docking, rather than predicting the overall structure based solely on the sequence. Although AlphaFold has outperformed all other methods so far, its efficiency makes it challenging for large-scale data. Additionally, while AlphaFold can predict the folded structure even for two proteins that cannot dock, EBMDock identifies the difficulty of docking by observing high energy levels after docking. It is worth noting that as the length of protein sequences increases, the speed of AlphaFold further slows down, and the accuracy noticeably decreases. However, our method primarily incurs increased memory usage, while other metrics remain relatively stable. We hope these points will ease your concern.

---

> ### Author Response · Authors · 2023-11-17
>
> > ***Q3: "In the case where ground truth is not available, we still need to use AF2 to predict the structure of each protein. As a result, the entire workflow for EBMDock won't be much faster than AFM."***
>
> The setting of our task is to predict complex structures from monomeric structures. While it is true that protein structure is determined by its sequence, in many cases, it is not always necessary to start with the sequence to predict the monomeric structure. Experimental techniques conducted in wet lab can also be employed to determine the structure, and the accuracy achieved through experimental methods undoubtedly surpasses that of prediction. In fact, the monomeric data used for training and testing in our study are derived from experimental measurements rather than predictions. Moreover, to the best of my knowledge, traditional docking software(HDock, etc), including those utilizing deep learning approaches(DIFFDOCK-PP, etc), typically initiate their predictions from monomeric structures, and these endeavors are indeed meaningful.
>
> > ***Q4: "Gradient of rotation $\nabla_RE(R,t)$ is not defined clearly， In practice, how do you calculate the gradient $\nabla_RE(R,t)$?"***
>
> We are sorry we didn't make it clear in the paper. We will explain it in detail below.
> In practice, we use the Euler Angle to characterize the rotation. Let $R=[\alpha, \beta, \gamma]^T$, the rotation matrix $R_M$ is calculated as follows:
> \begin{aligned}
> R_M &= R_{z}(\gamma)R_{y}(\beta)R_{x}(\alpha) \\\ &= \begin{pmatrix}
>         \cos(\gamma) & -\sin(\gamma) & 0 \\\        \sin(\gamma) & \cos(\gamma) & 0 \\\        0 & 0 & 1
>         \end{pmatrix}
>         \begin{pmatrix}
>         \cos(\beta) &  0 & \sin(\beta) \\\        0 & 1 & 0 \\\        -\sin(\beta) & 0 & \cos(\beta)
>         \end{pmatrix}
>         \begin{pmatrix}
>         1 & 0 & 0 \\\        0 & \cos(\alpha) & -\sin(\alpha) \\\        0 & \sin(\alpha) & \cos(\alpha)
>         \end{pmatrix}
> \end{aligned}
>
> The coordinates of the unbound ligand and receptor are $X_L$ and $X_R$, respectively. The docking pose is $(R_M X_L+t, X_R)$. For a residue pair $(li, rj)$ on the binding interface, the distance of them is $d_{i,j}= ||R_M X_{Li} + t -X_{Rj}||^2_2$. Then the energy can be calculated by equation (8) and (9) in the paper. After that, we can get $\nabla_{R}E(R,t)$ by backpropagation.
>
> > ***Q5: "Equiformer-PP architecture is not described."***
>
> We are sorry that we have not elaborated on the architecture of EquiformerPP. We add this part in Appendix E.1, see new pdf for details.

---

> ### Comment · Reviewer_XEYz · 2023-11-21
> **Thank you for your response**
>
> I would like to thank the authors for their response. I am concerned about the additional comparison with AFM for two reasons. First, the new test set only includes antibody-antigen complexes. It should include a diverse set of protein-protein interactions. It is pretty well known that AFM performs poorly on antibody-antigen complexes than other protein-protein complex types, so i unfair to just compare your EBMDock with AFM on antibodies. Second, reporting only the mean DockQ is also not fair. The protein-protein docking community also care about success rate, i.e., how many test cases with acceptable/medium/high docking quality (the acceptable quality threshold is DockQ > 0.23). If you look closely at the model performances, we see that AFM DockQ has a large standard deviation ($0.108 \pm 0.172$), which means some of them have a relatively high DockQ score. In contrast, the standard deviation for EBMDock is low ($0.05 \pm 0.02$). Therefore, it is likely that the success rate for EBMDock is nearly zero and much lower than AFM. Taking all these results into consideration, I would like to keep my original score.

---

> ### Author Response · Authors · 2023-11-22
>
> Thanks for your feedback. Regarding the two points you mentioned, here is our response.
>
> 1. Due to time and finance constraints, it is challenging to obtain results on AFM in the limited rebuttal window, thus we will include a diverse set of protein-protein interactions before camera ready.
>
> 2. We agree that AFM has a higher success rate compared to our approach (at the cost of much higher training overhead in terms of computing resources and data/labels, as well as inference overhead including computing resource and time latency -- see also some details in the below table). However, the larger variance also indicates that while AFM provides higher mean accuracy, it is relatively unstable. From this point of view, this instability can be compensated by utilizing our proposed method. Specifically, when visualization shows that the predicted results from AFM are completely irrational, we can use energy model to optimize them.
>
> From all of your feedback, you have been emphasizing the performance gap between our model and AFM, as well as the unfairness in comparing them, while admittedly overlooking our efforts and explanations about the different application scenarios. The table below provides a comprehensive comparison between our approach and AFM.
>
> | Methods | Efficiency                                  | Resource needed | Institute   | Senerio             | Training code  |
> | ------- | ------------------------------------------- | ---------------------------------------- | --------------------- | ---------------------------------------- | --- |
> | AFM     | low (6h for local AFM, 0.5h for ColabFold) | heavy (3TB disk space, multiple GPUs)* | giant company (DeepMind)  | structure prediction with high precision |  not available   |
> | EBMDock (ours) | High (8s for local)                         | lightweight (one RTX 3090 GPU)               | academic (university) | large-scale virtual screening            |    will be released |
>
> *the number is quoted from https://github.com/google-deepmind/alphafold

---

> ### Author Response · Authors · 2023-11-22
>
> Additionally, we would like to emphasize that we are not listing AFM here as a competitor that we have to outperform, but rather as a baseline since it is a well-known SOTA method. In fact, we explore new deep learning based methods aiming to significantly improve efficiency while maintaining a certain level of accuracy. Apart from us, there is also a series of works [1][2][3] making similar efforts after AFM. Even though all the performance may not match that of AlphaFold, our work has made progress and provided new insights for researchers.
>
> Besides, we can leverage AFM to further improve our energy model for docking. Specifically, protein sequence information is easily accessible and abundant, while structural information is scarce. Given sufficient time, we can leverage AF to predict complex structures and generate new training data to further expand our approach (the training set now contains 10,000 samples derived from experimental measurements). In this way, high-quality structure prediction using AFM can expand the training set, while inference on large-scale data using our method can still maintain high efficiency. We believe the existence of AFM should not negate the exploration of other new methods. As reviewer AgXx said, "All deep learning approaches for docking have failure modes in real-world applications, and exploring new ideas and being forthcoming about performance are both needed for the field to progress."
>
>
>
> [1] M Ketata, et al."DiffDock-PP: Rigid Protein-Protein Docking with Diffusion Models" ICLR 2023 MLDD
> [2] Y Luo, et al. "xTrimoDock: Rigid Protein Docking via Cross-Modal Representation Learning and Spectral Algorithm." bioRxiv (2023): 2023-02.
> [3] R Wang, et al. "Injecting Multimodal Information into Rigid Protein Docking via Bi-level Optimization" Neurips 2023

---

> ### Author Response · Authors · 2023-11-22
>
> **Underperforming industry-level AFM does not mean an academic reject.** You claimed that our work "is not going to be useful" because we can't match the performance of AlphaFold-Multimer (AFM) from giant company DeepMind. It is challenging for the academic community to compare with industry models in terms of accuracy due to limited resources. But we make significantly efforts in terms of efficiency. AFM is based on multiple sequence alignments (MSAs), leading to difficulty in improving the inference speed, and cannot be used for large-scale virtual screening (at least if given only limited budget). However, we construct a differential energy function and utilize an efficient sampling method to prevent exhuastive search (and also some space for future improvement as also appreciated by Reviewer AgXx, 94kF).
>
> Also, you said that "EBMDock needs to use AF2 to predict the structure of each protein in the case where ground truth is not available and the entire workflow won't be much faster than AFM". However, EBMDock as well as other docking methods [1,2,3,4,5] begin with structure are not designed for this case, we are designed for given monomers structures achieved from experimental techniques. It is biased to deny the significance of our efforts to improve efficiency with the case from another different setting.
>
> [1] Y Luo, et al. "xTrimoDock: Rigid Protein Docking via Cross-Modal Representation Learning and Spectral Algorithm." bioRxiv (2023): 2023-02.
>
> [2] M Ketata, et al. "DiffDock-PP: Rigid Protein-Protein Docking with Diffusion Models" ICLR 2023 MLDD.
>
> [3] R Wang, et al. "Injecting Multimodal Information into Rigid Protein Docking via Bi-level Optimization" Neurips 2023.
>
> [4] O, Ganea, et al. "Independent SE(3)-Equivariant Models for End-to-End Rigid Protein Docking" ICLR 2022
>
> [5] Y, Yan, et al. "The HDOCK server for integrated protein–protein docking" Nature Protocols 2020

---

> > ### Comment · Reviewer_XEYz · 2023-11-22
> > **Response**
> >
> > Thank you for your additional comments. I would like to clarify several things:
> >
> > 1) Regarding standard deviation, the point I want to make is that you need to report docking success rate, which is a standard practice in docking community.
> >
> > 2) I am totally fine for accepting a docking model that underperforms AFM but runs much faster. However, this model needs to have a decent performance. Having a DockQ score of 0.05 means (almost) none of your docked structures have acceptable quality (DockQ > 0.23). I am afraid that no biologists would use a model that never produces correct docking poses, even if it runs very fast. The concern here is not only underperforming AFM, but the absolute performance is too low to be useful in practice.
> >
> > 3) I am sorry to say that your model also underperforms HDOCK on the new test set, which is an "academic" docking software. Its DockQ is 0.090±0.187, which means it will have certain level of success rate. I suppose it's not great, but definitely better than EBMDock, which is almost zero.
> >
> > 4) There are other deep learning based docking models developed in academia with decent performance. For example, GeoDock [1] is a flexible protein-protein docking model whose average inference time is 0.76 second per instance, which is much faster than EBMDock (10.4 second) on DB5.5. I think it would be great to compare the success rate of EBMDock and GeoDock on DB5.5 test set.
> >
> > In summary, I will keep my decision for rejection due to the extremely low performance of EBMDock, and raise my confidence score from 3 to 4. I would like to thank the authors for their detailed response.
> >
> > [1] Chu et al., Flexible Protein-Protein Docking with a Multi-Track Iterative Transformer, 2023

---

> ### Author Response · Authors · 2023-11-22
>
> Thank you for your time and valuable feedback. Below is the detailed response.
>
> We apologize if you felt being offended by our previous response. There may be some misunderstandings since you have been emphasizing the gap between EBMDOCK and AFM. We may partially distort what you mean.
>
> 1. We used the data set collected from xTrimodock, the results of Hdock and AFM were quoted from xTrimodock since they were time-consuming to reproduce. They did not report success rate. If the pdf could be further updated, we will reproduce the experiment and report their success rates. Here we only provide the success rates of EBMDock, 1 / 68 for EBMDOCK(5) and 42 / 68 for EBMDOCK(5)-interface.
> 2. We fully understand your concern about our performance, which also need further efforts for the community. We admit that we do not perform well when blind docking, but for biologists, they can get a roughly surface knowledge when facing accurate docking (such as finding surface docking concave candidates). And in this case, our performance will increase extremely (42 / 68 success cases), which indicates practical and verifies the validity of our energy function. If our method is used to do large-scale virtual screening, it can also screen out the absolutely irrational structure even under low performance.
> 3. We only claim that we surpass HDock on CRMSD, and HDock do outperform EBMDock on dockQ. But it should also be noticed that the inference speed of HDock is relatively slow, almost 50x slower than EBMDock as well as other deep learning methods.
> 4. Thank you for proposing another baseline, since we did not come across their article during our research for this work. We will compare with them in the subsequent updates.
>
> In summary, we respect your valuable feedback and time, we will improve the paper for our final version based on your comments. Could you please kindly reconsider your rating if our feedback could ease some of your concerns (also seeing other reviewers comments).

---

### Official Review · Reviewer_S6B4 · 2023-11-01

**Soundness:** 3 good
**Presentation:** 4 excellent
**Contribution:** 3 good
**Rating:** 6
**Confidence:** 5

**Summary:**

This work proposes an energy-based model for rigid protein-protein docking.
The energy function is parameterized as an averaged negative distance log-likelihood between some residue pairs, which can be trained using standard contrastive divergence.
Represented as a SE(3) transformation (R, t) (6 degrees of freedom), a possible docking pose can be generated via Langevin dynamics guided by the energy function. In practice, the authors restrict the calculation of energy function to residue pairs on the binding site.
The experimental results show that the proposed method outperforms recent dl-based baselines and run much faster than traditional docking programs.

**Strengths:**

1. The paper is well presented with a straightforward idea, following the standard EBM paradigm.
2. A binding site prediction module is involved to restrict energy function parameterization on relevant residue pairs, which might be crucial for the success of the method.
3. The idea seems general and can be applied to other kinds of rigid-docking settings.

**Weaknesses:**

1. The method only considers the position of alpha carbon in each residue, which can be a crucial limitation. Being able to model the geometry of full backbone atoms or even sidechain atoms is necessary in real-world scenarios.

2. The authors include the DockQ score as one of their evaluation metrics. The original DockQ score includes the percentage of native contacts preserved in the predicted structure as one of criteria. However, the current method ignores all backbone atoms except for alpha carbon. The author should make this point clear.

3. The proposed method falls short of the AlphaFold-Multimer on both evaluation tasks. Although the authors have mentioned that the results of AFM might be optimistic (test set may overlap with the training set of the AFM), it would be better if they could report the results of both models on another benchmark set composed of proteins released after the cutoff date of the AFM's training set.

**Questions:**

1. Training an EBM with contrastive divergence where negative samples are generated by Langevin dynamics on the fly is quite time-consuming. May I know how much time it takes to train the EBMDock and have you tried any other techniques to reduce the cost?

2. Another straightforward idea of rigid Protein-protein docking is that we first train an inter-residue distance predictor and then recover the docking poses that best fit the predicted inter-residue distance map via gradient descent. Have you tried comparing EBMDock with such a model?

3. Apart from the diverse predictions, can you briefly illustrate the insights or motivation for using EBM to solve rigid protein-protein docking? (As training EBM using CD is so costly while training the diffdock-pp is relatively cheap.)

---

> ### Author Response · Authors · 2023-11-17
>
> Thank you for your time and valuable feedback. Below is the detailed response.
>
> > ***Q1: "The method only considers the position of alpha carbon in each residue, which can be a crucial limitation. Being able to model the geometry of full backbone atoms or even sidechain atoms is necessary in real-world scenarios."***
>
> Since internal structures of residues of the same class are relatively similar and modeling on residue level can be computationally efficient, we choose to use residue-level modeling in our experiment. Our method can be easily extended from residue level to atom level. To extend EBMDock to the atomic level, we only need to modify the nodes in the protein graph from representing a residue to representing an atom. Instead of learning the statistical potential of residue distances, we would learn the statistical potential of atomic distances. The rest of the network structure and training methods remain unchanged. Also, the challenge of protein-protein docking always lies in the prediction of the backbone structure, and we can use existing tools such as Rosetta to complete the side chain conformation. We hope these points will ease your concern.
>
> It should be noticed that most deep learning methods, such as Equidock and DIFFDOCK-PP(our baseline), are also based on modeling at the residue level.
>
> > ***Q2: "The authors include the DockQ score as one of their evaluation metrics. The original DockQ score includes the percentage of native contacts preserved in the predicted structure as one of the criteria. However, the current method ignores all backbone atoms except for alpha carbon. The author should make this point clear."***
>
> The original definition of DockQ indeed involves the use of all-atom representation, but here we substitute it with alpha-C and calculate it in the same manner for all baseline methods. This means our comparison is still fair although we used an alpha-C version of DockQ. Thanks for pointing this out, we have modified our manuscript to make this point clear.
>
>
> > ***Q3: "The proposed method falls short of the AlphaFold-Multimer on both evaluation tasks. Although the authors have mentioned that the results of AFM might be optimistic (the test set may overlap with the training set of the AFM), it would be better if they could report the results of both models on another benchmark set composed of proteins released after the cutoff date of the AFM's training set."***
>
>
> We select 68 antibody-antigen samples from Protein Data Bank which are released after October 2022 and have not been used to train AlphaFold-Multimer or as templates in HDOCK. Since the template library constructed by HDOCK may have encountered the test set mentioned in the article, we also include it here for a relatively fair comparison.
>
> | Method               | CRMSD ($\downarrow$) | DockQ ($\uparrow$) |
> | -------------------- | -------------------- | ------------------ |
> | HDOCK                |      15.779±6.364                |           0.090±0.187         |
> | AF-Multier           |        13.650±5.886              |             0.108±0.172       |
> | EquiDock             |     18.468±2.706                 |           0.043±0.017          |
> | DIFFDOCK-PP(5)       |   23.817±7.041       |   0.018±0.064           |
> | EBMDOCK(5)           |       14.870±4.930              |            0.054±0.022        |
> | EBMDOCK(5)-interface |      6.511±5.126              |           0.324±0.218       |
>
> The result here demonstrates that both HDOCK and AF-Multimer exhibit a significant decrease in performance on unseen cases. Although EBMDOCK's results also decline, they are close to those of AF-Multimer and surpass HDOCK in CRMSD. Additionally, when EBMDOCK is able to obtain rough docking interface information, the docking results are far superior to both AF-Multimer and HDOCK. We will add these results to our manuscript and hope they can ease your concerns.

---

> > ### Author Response · Authors · 2023-11-17
> >
> > > ***Q4: "Training an EBM with contrastive divergence where negative samples are generated by Langevin dynamics on the fly is quite time-consuming. May I know how much time it takes to train the EBMDock and have you tried any other techniques to reduce the cost?"***
> >
> > We take almost three days to train our EBMDock till converge on a machine with i9-10920X CPU, RTX 3090 GPU, and 128G RAM, which we think is acceptable. We have tried to use a fixed number of MCMC steps (typically fewer than required for convergence) to reduce the cost, since running Langevin MCMC till convergence to obtain an unbiased sample from $x \sim p_{\theta}(x)$ can be computationally expensive. We set the number of steps to 15.
> >
> > > ***Q5: "Another straightforward idea of rigid Protein-protein docking is that we first train an inter-residue distance predictor and then recover the docking poses that best fit the predicted inter-residue distance map via gradient descent. Have you tried comparing EBMDock with such a model?"***
> >
> > Yes, we have tried to predict the distance map of residues directly instead of using the Gaussian mixture model. However, directly fitting the distance has poor performance and the model lacks good generalizability.
> >
> > > ***Q6: "Apart from the diverse predictions, can you briefly illustrate the insights or motivation for using EBM to solve rigid protein-protein docking? (As training EBM using CD is so costly while training the diffdock-pp is relatively cheap.)"***
> >
> > We investigated the existing protein-protein methods and classified them into two categories. The traditional methods typically generate numerous complex candidates and utilize a scoring function to rank these candidates. As a result, they have good interpretability but are really time-consuming. The geometric deep learning methods treat the task as a regression problem. While these methods are efficient in speed, they are not stable and have limited performance. As a physics-based model, EBMDock combines their advantages, establishing a differentiable energy function with geometric deep learning to improve the performance and interpretability while ensuring efficiency with Langevin dynamics.
> >
> > By contrast, diffdock-pp can not provide the distribution over docking poses explicitly. The generated candidate poses have to be ranked by a confidence model, a classification network, the performance of which greatly influences the results. As for the training cost, diffdock-pp is much more expensive than EBMDock. Although the exact training time is not reported, according to their GitHub project, the model takes 170 epochs to converge on DIPS. Diffdock, a docking method for proteins and molecules, is the previous work of diffdock-pp. It is trained on four 48GB RTX A6000 GPUs for 850 epochs (around 18 days) [1].
> >
> > [1] Gabriele Corso, Hannes Stark, Bowen Jing, Regina Barzilay, and Tommi S. Jaakkola. Diffdock: ¨Diffusion steps, twists, and turns for molecular docking. In The Eleventh International Conference on Learning Representations, 2023.

---

> > > ### Author Response · Authors · 2023-11-22
> > >
> > > Dear Reviewer,
> > > Thank you for taking the time to review our paper. We greatly appreciate your insightful comments and suggestions, which have helped us improve the quality of our work.
> > >
> > > As the deadline for the discussion phase approaches, we have carefully considered your feedback and prepared a response to address the concerns raised. We have had discussions with the other three reviewers and have incorporated some updated content based on their feedback. Here we kindly request your feedback on our response and welcome any further questions or suggestions you may have.
> > >
> > > Best regards,
> > > The authors

---

### Author Response · Authors · 2023-11-17

We express our gratitude to all the reviewers for dedicating their time and providing valuable comments. They acknowledged that our work is well-motivated (AgXx)
, well-presented (S6B4, XEYz, 94kF), interesting (XEYz), provides a nice framework (94kF), and demonstrates a practical inference time (AgXx).

In the following response, we provide detailed answers to all the questions and comments point-by-point to strengthen our contributions further. We deeply appreciate the suggestions to improve this paper, and we have made corresponding modifications to the manuscript and have resubmitted a revised version.

If you have any further questions, please let us know so that we can provide a timely follow-up response.

---

### Author Response · Authors · 2023-11-20

Dear Reviewers,

We are deeply appreciative of the time and effort you have devoted to reviewing our paper. The insights you've shared have been crucial in helping us refine our work. As the discussion phase nears its conclusion, we truly hope that our detailed responses have met your expectations and assuaged any concerns. If there remain unresolved questions, we are more than willing to provide any necessary clarifications. Should our explanations bring clarity, it would be an honor for us if you might reconsider the evaluation of our work.

With heartfelt gratitude and warmest regards,

The Authors

---

### Author Response · Authors · 2023-11-20

Dear reviewers,

We would like to express our sincere gratitude again for your valuable comments and thoughtful suggestions. Throughout the rebuttal phase, we tried our best to address concerns, augment experiments and refine details in paper (a section was added in the appendix) with your constructive feedback. Since the discussion time window is very tight and is approaching its end, we truly hope that our responses have met your expectations and assuaged any concerns. We genuinely do not want to miss the opportunity to engage in further discussions with you, which we hope could contribute to a more comprehensive evaluation of our work. Should any lingering questions persist, we are more than willing to offer any necessary clarifications.

With heartfelt gratitude and warmest regards,

The Authors

---

### Meta-Review · Area_Chair_DDvy · 2023-12-14

**Metareview:**

This paper introduces a new energy-based model which can be used for estimating protein-protein docking poses (and energies). Overall the method is novel, and is a nice departure from other ML-based models. Three of the four reviewers raised their scores in response to discussions with the authors. However, it has been noted that overall performance remains poor in absolute terms; while this outperforms some other ML-based docking approaches (though not AlphaFold Multimer, though with the requirement for an MSA it is arguably not a fair comparison), overall it seems unlikely that this method would be applied as-is to large-scale virtual screening tasks without further refinement, or reliance on an external method to identify the interface.

**Justification For Why Not Higher Score:**

Very legitimate complaints about performance from one of the reviewers

**Justification For Why Not Lower Score:**

Other ML work in this area has issues as well; this is a new approach which arguably has more utility (as it learns an energy model)

---

### Decision · Program_Chairs · 2024-01-16

Accept (poster)